# MultiScan: Scalable RGBD scanning for 3D environments with articulated objects

**Yongsen Mao, Yiming Zhang, Hanxiao Jiang, Angel X. Chang, Manolis Savva**
Simon Fraser University
https://3dlg-hcvc.github.io/multiscan/

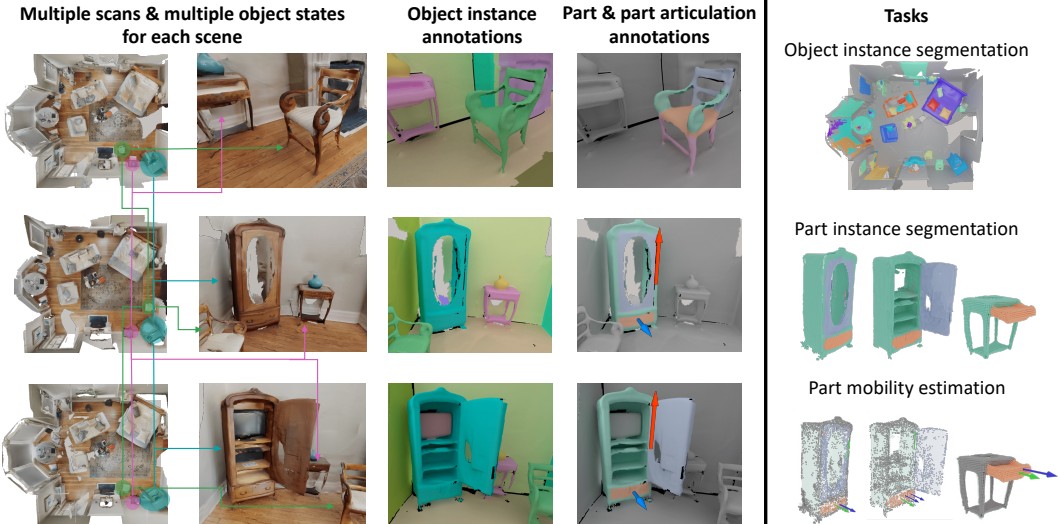

Figure 1: MultiScan offers 3D reconstructions of indoor scenes with multiple scans, capturing objects in multiple states, and with multiple levels of annotations. For each scene, we provide multiple scans (leftmost column) with objects in varied states (second column). Each scan is densely annotated with semantic object instances (third column) that are corresponded across scans. Each object is annotated with parts and part articulation parameters (fourth column). Different scans capture objects in different articulation states (e.g., the cabinet door is opened in the bottom row). The MultiScan pipeline and dataset enable investigation of holistic 3D scene understanding with articulated objects by supporting tasks that reason at multiple levels of a 3D scene (rightmost column).

## Abstract

We introduce MultiScan, a scalable RGBD dataset construction pipeline leveraging commodity mobile devices to scan indoor scenes with articulated objects and web-based semantic annotation interfaces to efficiently annotate object and part semantics and part mobility parameters. We use this pipeline to collect 273 scans of 117 indoor scenes containing 10957 objects and 5129 parts. The resulting MultiScan dataset provides RGBD streams with per-frame camera poses, textured 3D surface meshes, richly annotated part-level and object-level semantic labels, and part mobility parameters. We validate our dataset on instance segmentation and part mobility estimation tasks and benchmark methods for these tasks from prior work. Our experiments show that part segmentation and mobility estimation in real 3D scenes remain challenging despite recent progress in 3D object segmentation.

36th Conference on Neural Information Processing Systems (NeurIPS 2022).

# 1 Introduction

Datasets of 3D indoor environments are increasingly used for computer vision, robotics, and machine learning. ScanNet [7] and Replica [37] enabled much work on 3D semantic segmentation and object detection. Gibson [45] and Matterport3D [3] enabled work on vision-and-language, visual navigation, and many other embodied AI tasks. Even though the impact of such work is high, efforts are few and far between, typically limited by the availability of usable hardware and software infrastructure, and the significant time commitment required for data collection and annotation.

Moreover, these real-world 3D reconstruction datasets are not interactive. Reconstructions capture a static scene, providing a "frozen in time" snapshot that does not allow the kinds of rich interactions possible in the real world (e.g., opening and closing kitchen cabinetry). Thus, existing real scene datasets do not contain information about how objects can be rigidly moved and how object parts can articulate to open and close.

To address this challenge, we design MultiScan, a scalable 3D environment acquisition pipeline based on commodity devices (smartphones and tablets). Our pipeline allows for collection of and processing of raw RGBD data to produce 3D surface mesh reconstructions with textures. Rapid turnaround times for processing paired with a set of web-based annotation interfaces for semantic annotation at both object and part levels and annotation of part mobility parameters allow for efficient and scalable construction of articulated, real-world 3D scene datasets.

Using our software infrastructure, we collect a dataset of diverse real interiors each scanned at multiple points in time. Multiple scans per interior allow us to annotate and correspond objects that move rigidly and object parts that articulate. Unlike prior work on static 3D reconstruction datasets, we capture container interiors (cabinetry, fridges etc.) and collect multiple observations of the same objects and object parts in different states. These properties of the dataset allow us to carry out a dense multi-scale semantic annotation at the scene, object, and part levels.

We use the MultiScan dataset to systematically benchmark tasks at several levels (object segmentation, part segmentation and motion estimation) and evaluate the consistency of approaches for these tasks between scans of the same scene. We show that current methods for each of these tasks exhibit prevalent failure modes and leave much space for improvement. Moreover, consistency of predictions across observations of the same scene is low, showing that there is much work to be done to achieve consistent holistic 3D scene understanding.

In summary, we contribute: 1) a scalable real-world 3D environment acquisition and annotation pipeline; 2) a dataset of densely annotated 3D interiors with object, part, and part mobility annotations; 3) a systematic benchmark of methods for object and part segmentation and mobility estimation. We open-source our code and our data under the MIT license.

# 2 Related Work

We summarize related work on semantically annotated real-world scenes and interactive (i.e. articulated) objects. See Table 1 for a comparison against relevant prior work.

**Reconstructed 3D environments.** Existing reconstructed indoor environments are limited in multiple observations of the same scene, object part level and mobility information. SceneNN [13], ScanNet [7] and Replica [37] contain mostly single room scale reconstructions with dense semantic object annotations on mesh vertices. ARKitScenes [8] is the largest room-scale 3D dataset with 5,048 scans of 1,661 unique scenes but only 3D object bounding boxes are annotated. Matterport3D [3], Gibson [45], and HM3D [33] are all building-scale 3D datasets. Only Matterport3D [3] has manual semantic annotations at the object level. Armeni et al. [2] offer semi-automatic annotations based on MaskRCNN [11] for a limited number of categories in a subset of Gibson [34] scenes. Unlike our work, none of these datasets extend object level annotations to the object part and part motion levels.

The ScanNet [7] dataset contains several scans of the same environments. However, there was no control for whether objects are moved and no explicit mapping between object instances in different scans. RIO [41] and Rescan [9] both capture multiple scans of each environment with objects moved around and corresponded across scans. However, they do not capture or annotate articulated objects with different articulation states. To the best of our knowledge, our work provides the first dataset

| Dataset | Scenes | Scans | Objects | Parts | RGBD | Multi-state | Part articulations | Scene-obj-part hierarchy |
|---|---|---|---|---|---|---|---|---|
| SceneNN [13] | 100 | 100 | 1667 | ✗ | ✓ | ✗ | ✗ | ✗ |
| Replica [37] | 18 | 35 | 2843 | ✗ | ✗ | ✗ | ✗ | ✗ |
| ScanNet [7] | 707 | 1513 | 36213 | ✗ | ✓ | ✗ | ✗ | ✗ |
| Matterport3D [3] | 90 | 90 | 50851 | ✗ | ✓ | ✗ | ✗ | ✗ |
| ARKitScenes [8] | 1661 | 5048 | 67791 | ✗ | ✓ | ✗ | ✗ | ✗ |
| Rescan [9] | 13 | 45 | 1021 | ✗ | ✓ | ✓ | ✗ | ✗ |
| 3RScan [41] | 478 | 1482 | 43006 | ✗ | ✓ | ✓ | ✗ | ✗ |
| OPDReal [18] | — | — | 231 | 787 | ✓ | ✓ | ✓ | ✗ |
| AKB-48 [25] | — | — | 2037 | ≈4074 | ✗ | ✗ | ✓ | ✗ |
| MultiScan | 117 | 273 | 10957 | 5129 | ✓ | ✓ | ✓ | ✓ |

Table 1: Comparison of semantically-annotated real-world scene and articulated object datasets. MultiScan provides semantic annotations at both the object and part levels, unlike prior work. In addition, articulated parts are annotated with motion parameters and multiple scans capture the scene at different points in time with objects in varying articulation states (multi-state). Raw RGBD video sensor streams are also preserved. Unlike datasets that focus on single objects [18, 25], MultiScan objects are observed in scenes with realistic context, and annotated hierarchically (scene-obj-part).

capturing a scene with multiple scans at multiple points in time, with multiple states, and multiple levels of semantic annotation at the object, object part, and part motion levels.

**Interactive environments and objects.** Current methods for obtaining interactive environments suffer from a lack of scalability. Typically datasets with interactive scenes consist of synthetic assets designed by artists from scratch [22, 31, 48], or based on real world reconstructions [38]. Some prior work [10, 34] creates functionally-equivalent interactive scenes by basing the arrangement of objects on semantically annotated 3D reconstructions and replacing scanned objects with objects from existing synthetic datasets [42, 46]. Instead, we directly annotate articulations on the reconstructed objects, so that objects have more realistic and varied geometry and textured appearance.

The datasets of articulated objects that the above prior work draws upon are also limited in scale and quality, consisting mostly of synthetic 3D models [12, 42, 46, 49]. Martín-Martín et al. [28] created the RBO dataset consisting of 14 real-world articulated objects. More recently, larger datasets of real-world articulated objects were introduced with AKB-48 [25] focusing on small tabletop objects and OPDNet [18] on objects with openable parts. All three datasets are limited to isolated objects that are not arranged in scenes. We collect a dataset of 3D indoor environments with articulated objects. This allows for the study of holistic understanding of articulated objects in real scenes.

**Reconstruction of articulated objects.** Reconstruction of dynamically changing scenes is a challenging research problem. There is some recent work on reconstructing scenes in the presence of a few rigidly moving objects [44], but scenes with object articulations such as kitchen cabinetry remain challenging to reconstruct due to severe occlusions and partial observation in the presence of a manipulating human. Recently, there has been increasing interest in predicting part mobility from 3D meshes [12], single point clouds [23, 42, 49], multiple point clouds [35, 36], images [1, 18, 51], and depth sequences [15, 16]. Identifying movable parts and motion parameters for articulated objects that are statically reconstructed is a strategy for bypassing dynamic reconstruction. However, static reconstruction methods cannot directly handle multiple scans of the same object in different articulation states. To address this, flow-based methods to correspond points from multiple point clouds have been proposed [14, 50]. Other work jointly predicts motion and performs reconstruction by using implicit functions [20, 29, 43, 52] or neural radiance fields [39]. All the above work focuses on mobility prediction for isolated single articulated objects, and does not consider the placement of the objects in a realistic 3D environment. Moreover, these recent advances were enabled in part by the availability of articulated object datasets [42, 46]. As a step toward encouraging more work in this direction at the scene level, we provide the first dataset of articulated real-world scenes.

# 3   Approach

We follow an approach similar to the ScanNet [7] data collection pipeline for acquiring and semantically annotating 3D reconstructions of indoor environments. Key differences are that our framework produces: 1) textured meshes with dense semantic annotations on mesh triangles (vs only at vertex

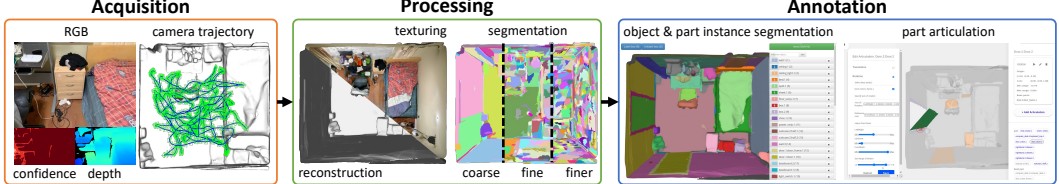

Figure 2: Overview of the MultiScan pipeline. During **acquisition** we use mobile devices to collect RGBD sequences and camera trajectories in a given scene. Then, in the **processing** stage we reconstruct aligned textured 3D meshes and compute an unsupervised multi-scale segmentation that will be used for annotation. During **annotation** we use a set of web-based interfaces to semantically annotate object and part instances in each scan, and part articulation parameters.

points); 2) multi-level hierarchical annotations of object instances and parts; 3) object articulation and motion annotations; and 4) correspondences between instances in scans of the same scene. Our pipeline also provides data collection apps for iOS and Android commodity devices.

Figure 2 illustrates the MultiScan reconstruction and annotation pipeline. The iOS and Android scanning apps we developed use built-in LiDAR sensors, stereo camera pairs, or single RGB cameras with monocular depth estimation (Section 3.1). This allows us to acquire scans with a variety of devices and sensor types. We then carry out RGBD fusion to produce the 3D mesh geometry and compute textures for the geometry using the aligned RGBD image sequence (Section 3.2). We semantically annotate object instances in each scan, correlating instances across scans of the same scene, and specifying articulated parts and their motion parameters (Section 3.3).

## 3.1 Acquisition

We developed scanning applications for iOS and Android mobile devices. These applications capture RGB, depth, depth confidence and IMU sensor streams, as well as camera pose trajectories. An associated UI allows the operator to specify metadata (scene description etc.), and to upload the scan to a server for reconstruction (see supplement for details). A web interface is also available to browse all uploaded data and check reconstruction quality to determine if scanning again is required.

To create the dataset in this paper, five volunteers used the iOS scanning app with three iPad Pro 2020 and two iPhone12 Pro devices to collect scans from residences, schools, and public spaces in several geographic locations in Canada, the United States, and China. Consent was obtained from the volunteers and from space owners for any non-public spaces that were scanned. No people or personally identifiable information was scanned. During scanning, users move freely through the scene with the device in hand and record an RGBD video. Each scene is scanned multiple times after objects and their parts are moved to different configurations. During the first scan, most of the articulated objects (e.g. drawers and cabinets) are left in a closed state. For additional scans, we open parts such as drawers and cabinets, and reposition objects that are commonly moved during daily use (e.g., chairs, pillows, kitchen utensils) to obtain scans of the scene in a different state.

Collected data are captured at 60 Hz with synchronized device timestamps allowing for alignment between data modalities. RGB video frames have a resolution of 1920x1440, while depth and depth confidence frames are 256x192. In total, we captured about 4.19M frames (19.4 hours) of RGBD video data. The median scan duration is about 15.3K frames (4.3 minutes).

## 3.2 Processing

We fuse depth frames into a Truncated Signed Distance Field (TSDF) volume using the camera trajectory captured from the device, and extract a surface mesh reconstruction using the marching cubes algorithm [26] from Open3D [53]. We use a voxel size of 9.77 mm, and a truncation value of 0.08m. We simplify the surface mesh using InstantMeshes [17] to reduce the total number of vertices and faces. Then, we clean the mesh using MeshLab [6] to remove isolated components and degenerate faces, and to align scans of each scene to a common coordinate frame. We then generate textures for the surface mesh using the multi-view stereo texturing approach of Waechter

et al. [40]. The textured mesh preserves high frequency details on the surface geometry, allowing us to distinguish and annotate cabinetry, plates etc. that would otherwise be merged into other surfaces.

To enable efficient annotation, the resulting mesh is segmented using an unsupervised hierarchical graph-based segmentation algorithm as employed by ScanNet [7], treating the triangle mesh as a connected graph with mesh vertices (or faces) at the nodes, and mesh edges (or face-face adjacency edges) at the edges. We create a coarse and fine segmentation. The coarse segmentation uses only mesh vertex normal or face normal differences to compute edge weights for the graph cut algorithm. The fine segmentation also includes RGB colors to break coarse clusters based on surface appearance.

The processing steps above are done automatically for each uploaded scan on a processing server (Intel i9-10900F CPU, 32GB RAM, Nvidia RTX 3090Ti GPU). The median processing time per scan is approximately 8 minutes, with initial surface reconstructions typically being available for viewing on a web portal within about a minute. The relatively fast turnaround time enables repeated scanning at each location, and scalability of the pipeline with commodity devices and novice operators.

### 3.3 Annotation

Our annotation framework consists of several phases: 1) reconstruction quality check and scan alignment; 2) semantic annotation of objects and parts; 3) motion annotation; and 4) verification. We densely annotate objects in the scene with object and part labels, also defining a semantic front and up direction for each object to produce semantically-oriented bounding boxes (OBBs) that give the pose of the object in each scene. Object and part instances are correlated across scans of the same scene with consistent object and part IDs. Unlike prior work on semantic annotation tools for 3D reconstructions [7], our pipeline allows for fine-grained triangle-based annotation in addition to pre-segmented clusters. Our motion parameter annotation tools allow for annotating semantically meaningful motion ranges (consistently defined opened and closed state values in a motion range), unlike related prior work [46, 47]. Here, we provide a high-level overview of the object and part instance annotations, and motion parameter annotations. See the supplement for more implementation details of the complete annotation pipeline.

**Object and part instances.** We use a hierarchical labeling interface to paint object and part instances. Annotators specify part level annotations for objects that can be articulated with labels of the form `object_id:part_id = object_category.object_index:part_category.part_index` identifying the object and part category and instance. Corresponding objects and parts between scans are specified with the same `object_id` and `part_id`. We also annotate semantically-aligned oriented bounding boxes (OBBs) for each object (with consistent "front" and "up" directions defined for each object).

**Motion parameters.** Using the part instance annotations, we segment each object into parts to create a part connectivity graph to define a kinematic chain for the articulated parts. Then, annotators indicate movable parts, fixed parts, movable objects, fixed objects, and architecture elements (e.g., walls). For each movable part, the annotator specifies motion parameters: the base part (selected from the connected parts), motion type (translation or rotation), axis of motion, motion origin (for rotation), and motion range.

## 4 Dataset

In this section, we describe the MultiScan dataset and provide summary statistics of the scene, object, and part level annotations. MultiScan includes 273 scans of 117 indoor scenes across 12 scene types, with at least 2 scans per scene for 101 scenes.[1] The spaces captured in the MultiScan dataset are mostly single rooms with the size of a room ranging from a small bathroom or laundry room to large living rooms and library rooms. In rescans of the same scene, the states of articulated objects and the positions of movable objects are changed. We define an articulated object to be an object consisting of rigid parts that are connected by joints, and a movable object to be an object that is not attached to the architecture, and is movable without using specialized tools.

Overall, 10957 total object instances were annotated, of which 2215 are architectural elements (wall, ceiling, floor), 8742 are non-architectural objects, and 982 are articulated objects. We annotated a total of 5129 parts, out of which 3049 were moving parts (2113 exhibiting rotation, and 936 translation).

---

[1]Some rescans did not result in acceptable reconstructions.

| Textured mesh reconstruction | Part instances | Semantic OBBs | Part articulations |
| --- | --- | --- | --- |

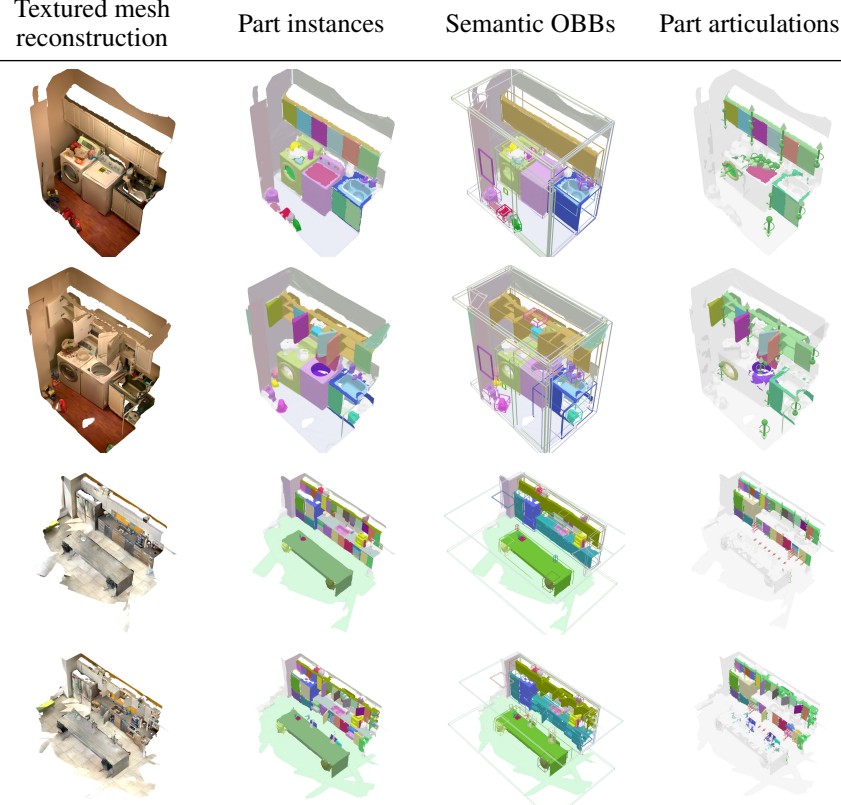

Figure 3: Annotated scans from the MultiScan dataset. Each scan is reconstructed as a textured mesh, which is then annotated with object and part semantic instances, object-level semantic oriented bounding boxes (OBBs), and part articulations.

During annotation, annotators used 419 fine-grained category names that were later grouped into coarser labels for our benchmark tasks. The supplement provides more statistics including the distributions over room categories, object and part categories, and number of parts per object.

The scans were annotated by 8 student volunteers. Annotation took on average 1h and 22m per scan with the bulk being spent on semantic segmentation (approximately 1hr and 12min for segmentation, 9min for articulation, and 1min for semantic OBBs). Verification was performed by the authors (∼10min per scan, with 6min for segmentation, 3min for articulation, and 1min for OBB verification). Figure 3 shows example annotated scans. Please see the supplement for more examples and statistics.

## 5    Experiments

We showcase the value of our dataset on the following tasks: 3D instance segmentation of objects and parts, and articulated object mobility prediction, evaluating recent methods for each task. We also introduce and measure object detection consistency across multiple scans for the same scene. We include results and examples for the val set in the main paper, and the test set in the supplement.

**Task dataset preparation.** We benchmark several point-cloud based methods on our MultiScan dataset. To get per-point semantic labels from our triangle based annotation, we assign the same semantic label to the three vertices of each triangle. We select the top 17 most common object categories, excluding room architecture (floor, wall, beam, etc.) and small objects (bounding box volume below $0.008\ m^3$, slightly smaller than a pillow). For part segmentation and mobility prediction, we predict articulated parts from objects extracted from a scene, and use either the ground truth or predicted object segmentation. To ensure that extracted objects have sufficient point samples, if an object has fewer than 4096 points we sample additional points from the surface mesh distributed

Table 2: Object instance segmentation results on the val set. Values are mean AP scores. All methods have fairly low scores for most categories, except chairs and toilets which are frequent and distinctive objects. SSTNet has the best overall performance, followed by HAIS and PointGroup.

| | all | door | table | chair | cab | win | sofa | pillow | tv | curt | bin | sink | bpk | bed | fridge | toilet |
|---|---|---|---|---|---|---|---|---|---|---|---|---|---|---|---|---|
| PG [19] | 26.2 | **35.9** | 22.0 | 65.8 | 16.0 | **10.3** | 29.2 | 10.8 | 24.7 | **2.6** | 32.8 | 43.9 | 1.4 | 60.5 | **16.8** | 64.3 |
| SSTNet [24] | **32.6** | 30.2 | **28.5** | **69.5** | 18.8 | 8.6 | **50.6** | **18.8** | 38.4 | 0.7 | **42.9** | 44.5 | **15.8** | **66.4** | 11.7 | **83.1** |
| HAIS [4] | 30.1 | 32.3 | 27.9 | 67.3 | **20.6** | 7.5 | 37.1 | 13.4 | **40.7** | 0.4 | 32.5 | **52.7** | 8.1 | 59.4 | 13.9 | 72.9 |

Table 3: Part instance segmentation results on the val set. Values are AP scores using ground-truth (left) and predicted (right) object instance segmentation. When using predicted objects, we use the same method for both object and part segmentation.

| | Ground-truth segmentation | | | | | | Predicted segmentation | | | | | |
|---|---|---|---|---|---|---|---|---|---|---|---|---|
| | all | static | door | drwr | win | lid | all | static | door | drwr | win | lid |
| PG [19] | 24.8 | 56.5 | 26.5 | 4.8 | **0.1** | 36.0 | 8.2 | **9.7** | 8.8 | 0.3 | 0.0 | 22.2 |
| SSTNet [24] | **29.8** | 53.4 | **35.0** | **12.0** | **0.1** | **48.4** | **9.5** | 8.5 | 6.6 | 0.8 | 0.0 | **31.6** |
| HAIS [4] | 24.6 | **58.2** | 23.3 | 8.1 | 0.0 | 33.4 | 9.1 | 8.3 | **9.4** | **1.8** | 0.0 | 26.3 |

uniformly based on triangle area. To allow for reproducible benchmarks, we split the data into approximately 70/15/15% train/val/test sets partitioned by scene (see supplement for statistics).

## 5.1 Object and part instance segmentation

We compare three methods with good performance on the ScanNet benchmark [7]: PointGroup [19], SSTNet [24], and HAIS [4]. All three methods rely on a SparseConv UNet backbone network for extracting per-point features, that are then clustered and scored. SSTNet [24] and HAIS [4] build on PointGroup [19] with hierarchical grouping of points into object instances. Object segmentation is done with the 17 most common categories, while part segmentation is done with common moving part categories (door, drawer, window, lid) and the static base part.

**Implementation details.** For each method, we apply the same settings for both object and part segmentation experiments. In our experiments, we use point clouds with position (xyz) and color (rgb) information. For SSTNet [24], we train using the original implementations with a learning rate of 1e-3, and AdamW [27] as the optimizer. For PointGroup [19] and HAIS [4], we train using a re-implementation[2] with Minkowski Engine [5] as the backbone with a learning rate of 1.5e-3, and Adam [21] as the optimizer. We apply data augmentation during training for all three methods: random jitter and mirror in the horizontal plane. We train all methods on the MultiScan train split for 512 epochs. We set batch size 4 and 64 for object and part segmentation, respectively.

**Metrics.** We report $AP$, the average of mean Average Precision over different IoU thresholds from 0.5 to 0.95. See the supplement for the AP scores at IoU thresholds of 0.25 and 0.5 ($AP_{25}$ and $AP_{50}$).

**Results.** We report the performance on object (Table 2) and part instance segmentation (Table 3). Mean values for these and other results are computed over three seeds. See the supplement for complete tables also including the standard error on the mean values. For object-level segmentation, we see a similar trend as the ScanNet benchmark [7] with SSTNet [24] and HAIS [4] performing better than PointGroup [19]. Note we omit from Table 2 categories (suitcase, microwave) that appear rarely (<2 instances), as all methods fail to predict these objects. We also observe all methods have better performance on large and typical-looking objects with few appearance variations (e.g., toilets, beds). In contrast, all methods struggle on poorly reconstructed objects such as windows and curtains, and small objects such as backpacks, pillows and suitcases. Refrigerators and microwaves are also challenging as they are often embedded in cabinets and poorly reconstructed. Figure 4 shows example object instance predictions. SSTNet has the cleanest segmentation results, but still fails to identify all the separate cabinets.

For part-level segmentation, we first extract the points for the object from the scene mesh and then center and align the points. We experiment with using both ground-truth object instances as well

---

[2] https://github.com/3dlg-hcvc/minsu3d

Table 4: Breakdown of part instance segmentation for open and closed parts on the val set. Overall, all three methods perform better for open parts. The door and drawer openable parts are easier to detect when opened, but window are easier when closed. Windows are likely challenging opened as the window surface would not be there (e.g., parallel sliding windows).

| | Opened parts | | | | | Closed parts | | | | |
|---|---|---|---|---|---|---|---|---|---|---|
| | **all** | door | drwr | win | lid | **all** | door | drwr | win | lid |
| PG [19] | 27.2 | **59.4** | 20.5 | **0.1** | 28.5 | 13.8 | 18.6 | 0.5 | 0.2 | 36.0 |
| SSTNet [24] | 28.2 | 34.7 | 28.3 | 0.0 | **49.9** | **19.1** | **29.8** | **5.1** | **0.5** | **41.2** |
| HAIS [4] | **29.5** | 53.7 | **35.6** | 0.0 | 28.8 | 12.1 | 15.0 | 0.4 | 0.1 | 32.8 |

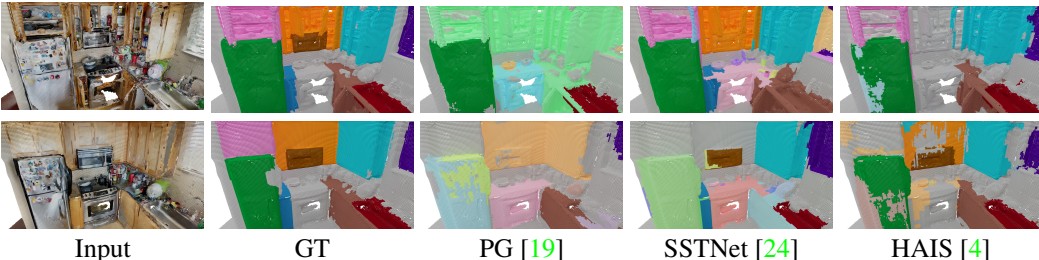

|   Input   |   GT   |   PG [19]   |   SSTNet [24]   |   HAIS [4]   |

Figure 4: Object instance segmentations on val set. We show two scans of the same scene with objects in different states. Objects with color matching the ground-truth (GT) are correctly predicted instances. Objects with different colors indicate incorrect semantic labels. We see that object instance segmentation is challenging for real articulated scenes, especially for the cabinets at the top.

as predicted object instances. In both cases, we align and center the extracted points using the ground-truth semantic up/front directions. From Table 3, we see that part-level segmentation can be quite challenging. We hypothesize this is due to small parts and fine-grained details (e.g., thin edges of doors, or door handles). With ground-truth object segmentation, the methods do a fair job at predicting the static part but are less good at predicting moving parts. All methods perform quite badly when using predicted object segmentations. In particular, static part segmentation drops considerably with predicted segmentations. Table 4 shows that door and drawer are more challenging in the closed state that in the open state, while the opposite holds for window. In the closed state, it is challenging to identify boundaries between adjacent doors (see Figure 5) and identify the static part, which is largely occluded. These results indicate that part segmentation at the scene level would be quite challenging. In initial experiments for direct part segmentation at the scene level (without predicting object instances first), we saw even poorer results.

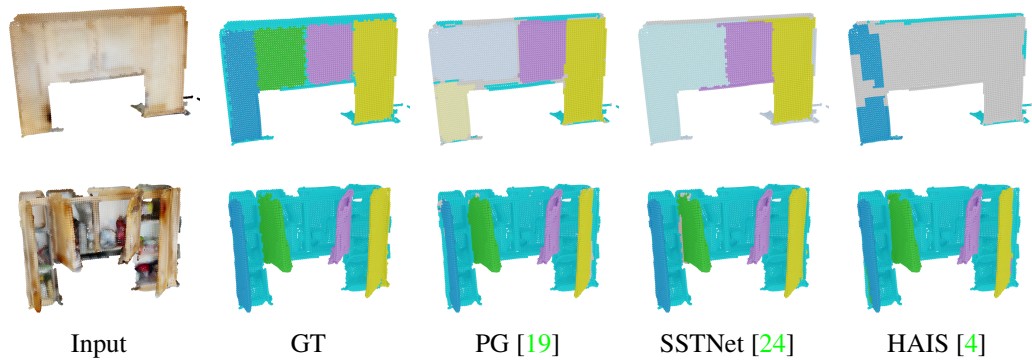

|   Input   |   GT   |   PG [19]   |   SSTNet [24]   |   HAIS [4]   |

Figure 5: Part instance segmentation for two states of a kitchen cabinet unit from the val set. Note that it is challenging to segment the boundary of the cabinet doors in the closed state (top). The static part (interior) is also largely unobserved in the closed state.

Table 5: Object instance segmentation consistency on val set. HAIS provides the most consistent object segmentations across scans of the same scene. Bathroom and office scenes are overall easiest, while bedroom and kitchen scenes are quite challenging likely due to the prevalence of hard-to-segment cabinetry.

| | **all** | balc | bathroom | bed | bks/lib | dining | hall | kitchen | living | lobby | misc | office |
|---|---|---|---|---|---|---|---|---|---|---|---|---|
| PG [19] | 44.43 | **29.63** | 73.61 | 30.97 | 32.94 | **39.39** | **44.44** | 39.93 | 25.00 | 63.33 | **40.74** | **68.75** |
| SSTNet [24] | 40.91 | 18.52 | 62.50 | **37.45** | **40.08** | 33.33 | 41.67 | 29.24 | **41.67** | 63.33 | 25.93 | 56.25 |
| HAIS [4] | **44.57** | 22.22 | **81.94** | 36.16 | 36.11 | 36.36 | 33.33 | **40.88** | 33.33 | **66.67** | 40.74 | 62.50 |

Table 6: Mobility estimation on the val set comparing OPDPN [18] and Shape2Motion [42] on objects extracted using ground-truth (GT) and predicted segmentations from SSTNet [24].

| seg | method | IoU↑ | EPE↓ | MD↓ | OE↓ | TA↑ | Movable part R | Movable part P | Movable part F1 | Motion type R | Motion type P | Motion type F1 | Motion+Axis R | Motion+Axis P | Motion+Axis F1 | Motion+Axis+Origin R | Motion+Axis+Origin P | Motion+Axis+Origin F1 |
|---|---|---|---|---|---|---|---|---|---|---|---|---|---|---|---|---|---|---|
| GT | OPDPN | 54.80 | 0.96 | 0.48 | 3.99 | 92.59 | 1.59 | 3.09 | 2.08 | 1.28 | 2.56 | 1.69 | 1.21 | 2.33 | 1.58 | 0.83 | 1.47 | 1.06 |
| GT | S2M | **70.09** | **0.62** | **0.31** | 1.03 | 94.31 | **15.50** | 12.37 | 13.76 | **14.06** | 7.53 | **9.79** | **13.98** | 7.49 | **9.74** | **10.28** | 5.50 | **7.15** |
| GT | S2M (aug) | 67.69 | 0.93 | 0.49 | **0.72** | **96.47** | 13.45 | **13.80** | **13.58** | 9.14 | **8.65** | 8.88 | 9.14 | **8.65** | 8.88 | 5.97 | **5.66** | 5.81 |
| SSTNet | S2M | **70.50** | 0.59 | 0.29 | 1.41 | 94.85 | 9.37 | 17.02 | 12.09 | 8.69 | 11.03 | 9.68 | 8.62 | 10.94 | 9.59 | 6.05 | 7.64 | 6.72 |
| SSTNet | S2M (aug) | 69.22 | 1.04 | 0.52 | **0.57** | 95.95 | 6.73 | 15.68 | 9.40 | 4.53 | 10.35 | 6.29 | 4.53 | 10.35 | 6.29 | 2.87 | 6.47 | 3.97 |

**Segmentation consistency.** We also evaluate the consistency of segmentation across scans of the same scene. For a scene with $K$ scans, we define $C_{scene}(S)$, the consistency score for a scene $S$, to be the average consistency between each pair of scans $(s_i, s_j)$ for the scene. The overall consistency score C is then the average over the scenes. To compute $C_{scan}(s_i, s_j)$, we consider the $M_{ij}$ objects that are common between two scans, and evaluate whether each ground truth object instance $o_m$ has an instance segmentation prediction. For each object $o_m$, we check if $o_m$ has a prediction in both $s_i$ and $s_j$. We match predicted objects to ground truth by taking the predicted object with the highest IoU > 0.5 for each ground truth object. We enforce that at most one predicted object is matched to each ground truth object. The consistency score of the scan is the average over the $M_{ij}$ objects in the scene that are shared between the two scans. We report the object segmentation consistency across scans in Table 5. We note that HAIS [4] gives the highest consistency scores.

## 5.2 Mobility Prediction

As a key feature of MultiScan is the annotation of motion parameters for real-world objects, we benchmark methods for part mobility estimation. We compare two category-agnostic methods that operate on point clouds: Shape2Motion (S2M) [42] and OPDPN [18]. Both use a PointNet++ [32] backbone. Note that unlike the part instance segmentation task, we do not differentiate between part semantic labels (i.e. a part is either `moving` or `static`). Given an input point cloud $Q = \{q_i \in \mathbb{R}^3\}_{i=1}^N$, we predict the part for each point, and the motion parameters for each part, by outputting the set of moving parts $P = \{p_k\}_{k=1}^K$, and their joint parameters $J = \{j_k\}_{k=1}^K$. We assume each moving part $p_k$ is associated with a single joint $j_k = \langle t_k, a_k, o_k, \rangle$ with joint type $t_k = \{\texttt{rotation}, \texttt{translation}\}$, joint axis direction $a_k \in \mathbb{R}^3$, and joint origin $o_k \in \mathbb{R}^3$.

**Implementation details.** We use the released implementation of OPDPN and train using the Adam optimizer with learning rate of 0.001, batch size 32 for 600 epochs. We reimplement S2M in PyTorch. Compared to the original implementation, our re-implementation is faster and easier to run, and gives comparable performance (see supplement). We train MPN (1st stage of S2M) with batch size 16 for 800 epochs, PMM (2nd stage) with batch size 32 for 200 epochs. and MON (3rd stage) with batch size 16 for 200 epochs. All stages use Adam [21] with a learning rate of 0.001. As data augmentation for point positions we jitter and mirror in the horizontal plane, and for point color channels shift randomly by a sample drawn from $\mathcal{N}(0, 0.0025)$ (RGB channels are normalized to [-1,1]).

**Metrics.** We evaluate the mobility predictions using two sets of metrics. Following Shape2Motion [42], we report mean IoU, Endpoint Error (EPE), Minimum Distance (MD), Orientation Error (OE), and Motion Type Accuracy (TA) for matched parts. IoU measures the quality of the movable part segmentation, while the other metrics measure motion parameter accuracy. EPE [30] is used for 3D motion flow and measures the average distance between points of the moving part (we take the ground-truth part segmentation) as the part is moved according to the predicted motion vs the ground-truth motion (for rotation, the part is rotated to $180°$, and for translation, the part is moved by 1m). MD and OE measure the minimum distance and the angle between the ground-truth axis

Table 7: Comparison of Shape2Motion [42] performance on open vs closed parts evaluated on the val set with ground truth (GT) object instances.

| State | IoU↑ | EPE↓ | MD↓ | OE↓ | TA↑ | Movable part | | | Motion type | | | Motion+Axis | | | Motion+Axis+Origin | | |
|---|---|---|---|---|---|---|---|---|---|---|---|---|---|---|---|---|---|
| | | | | | | R | P | F1 | R | P | F1 | R | P | F1 | R | P | F1 |
| Closed | **67.76** | 1.00 | 0.56 | 0.81 | 95.73 | **14.16** | **14.58** | **14.34** | **8.18** | **7.77** | **7.93** | **8.18** | **7.77** | **7.93** | **5.07** | **4.82** | **4.92** |
| Opened | 66.52 | **0.72** | **0.36** | **0.45** | **100.00** | 12.51 | 9.57 | 10.77 | 6.26 | 3.74 | 4.67 | 6.26 | 3.74 | 4.67 | 4.75 | 2.83 | 3.54 |

direction and the predicted motion axis. Note this set of metrics is computed over the matched parts (with IOU>0.5). EPE, OE, and MD are all computed when the motion type is predicted correctly, and MD is only computed for rotation (as only the direction is relevant for translational motions). Because this set of metrics is only for matched parts it does not measure how many ground truth parts are predicted or how many predicted parts are valid movable parts. Thus, we also report the recall (R), precision (P), and F1 for predicting if a part is movable, the motion type, and whether it correctly predicted the motion axis direction (OE< 10°) and origin (MD<0.25 of the diagonal of the object).

**Results.** We report the performance of OPDPN and S2M on extracted objects with ground truth and predicted object segmentation (we use the best performing model SSTNet) in Table 6. Compared to the synthetic S2M dataset, our MultiScan dataset is considerably more challenging due to partial and noisy reconstructions. On the MultiScan dataset, we also experiment with different input features and data augmentation (see supplement). Our experiments in the main paper are with position, color and normal. The much simpler OPDPN performs surprisingly well compared to S2M for the matched parts. However, when we consider the recall, precision, and F1 metrics, we find that the performance of OPDPN is considerably worse. We also compare the mobility prediction performance for open and closed parts (see Table 7). See the supplement for qualitative examples.

## 6 Conclusion

**Limitations.** Though our focus was on scalable articulated scene acquisition we had access to a limited number of spaces. Thus, our data exhibits biases in the types of scenes and types of objects observed and does not capture the full diversity of real interiors. This may lead to negative societal impacts due to under-representation of spaces for communities that are not geographically close to us. Moreover, our reconstruction pipeline is limited by geometric resolution constraints making it challenging to represent and annotate smaller parts and objects. Also, our experiments only evaluate point-cloud based methods for segmentation and mobility estimation. Experiments with alternative methods and settings may reveal different trends, and are an interesting direction for future work.

We described the design and implementation of MultiScan: a scalable articulated RGBD scene dataset construction pipeline. We used this pipeline to scan interiors with multiple states of articulated and rigidly-movable objects. We richly annotated the resulting dataset with multiple levels of semantic and motion parameter information, at the scene, object, and part levels. Using this dataset, we benchmarked methods for object and part instance segmentation, and part mobility estimation. We found that segmentation and motion parameter estimation for articulated objects remain challenging with much space for improvement. We believe MultiScan will catalyze work on holistic understanding of interactive 3D environments and enable progress in perception for embodied AI and robotics.

**Acknowledgements.** This work was funded in part by a Canada CIFAR AI Chair, a Canada Research Chair, NSERC Discovery Grant, a research grant by Facebook AI Research, and enabled by support from WestGrid and Compute Canada. The iOS and Android scanning apps were developed by Zheren Xiao and Henry Fang. We thank Qirui Wu for his help in re-implementing PointGroup with the Minkowski engine and contributions to the minsu3D code repository. We also thank Zhen (Colin) Li for collecting additional scans, and Henry Fang, Armin Kavian, Han-Hung Lee, Zhen (Colin) Li, Weijie (Lewis) Lin, Sonia Raychaudhuri, and Akshit Sharma for helping to annotate data.

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
