# MultiScan: Scalable RGBD scanning for 3D environments with articulated objects Supplemental Materials

**Yongsen Mao, Yiming Zhang, Hanxiao Jiang, Angel X. Chang, Manolis Savva**
Simon Fraser University
https://3dlg-hcvc.github.io/multiscan/

This supplemental document provides the following additional contents to support the main paper:

## A  MultiScan pipeline details

In this section we describe in more detail the different phases of the MultiScan pipeline: acquisition, reconstruction, and annotation.

### A.1  Acquisition details

We developed scanning apps for both iOS and Android mobile devices. The scanning app captures RGB, depth, confidence streams, camera poses, and provides UI for the user to specify metadata (scene type, location, etc) and upload the scans for a server for reconstruction. Users move freely through indoor scenes with the devices in hand, and record RGBD scans of the environment.

Owners of non-public spaces that were scanned provided their consent by reading and agreeing to the terms below:

```
I agree to allow Simon Fraser University to use data of my space for academic and/or non-
commercial research purposes as described by the Creative Commons Attribution-NonCommercial
4.0 (CC BY-NC 4.0) license (https://creativecommons.org/licenses/by-nc/4.0/).

How will the data be used?
The data will be used for academic research, such as 3D computer vision and artificial
intelligence. With your permission the data from your space will become part of a collection
of spaces around the world.

Is it anonymous? Is any of my information distributed in the data?
Yes, it is anonymous. We will not distribute any information about your name, address, or
other personal information. However, please make sure that there is no personal information
that is visible when we collect the data (for example pieces of paper with personal
information, or photos). If there is such information and it cannot be hidden, please let us
know so that we avoid taking pictures of it.
```

The iOS app uses Apples's ARKit library and the Android app uses Google's ARCore library. Active sensors such as time-of-flight (ToF) and LiDAR sensors are detected and used. If unavailable, less accurate estimated depth frames are acquired. Recorded data streams are compressed with the H.264 codec for RGB video, and zlib compression for other data streams such as depth and depth confidence. Associated 6 DoF camera poses and timestamps for each frame are also stored in json line format. See Figure 1 for screenshots showing different parts of the acquisition app UI from both iOS and Android devices.

36th Conference on Neural Information Processing Systems (NeurIPS 2022).

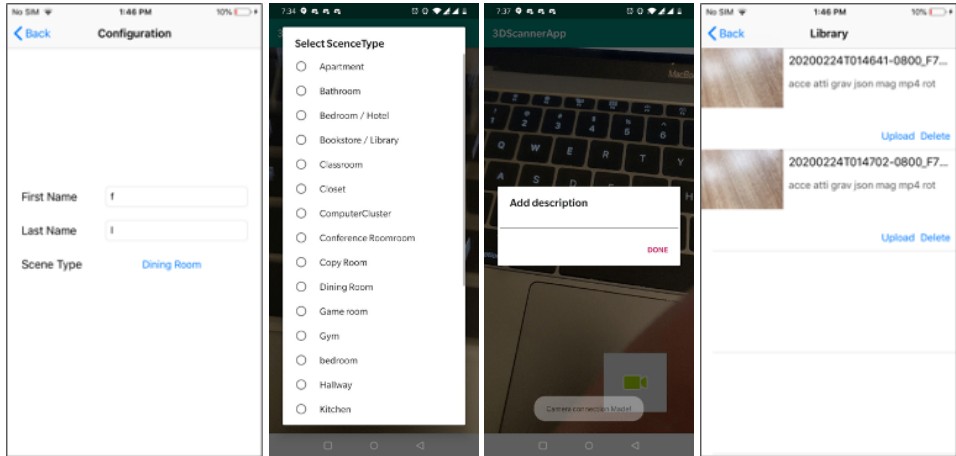

Figure 1: Screenshots from the iOS and Android acquisition apps showing parts of the interface allowing the user to specify scene metadata and description text, list acquired scans and upload to a processing server.

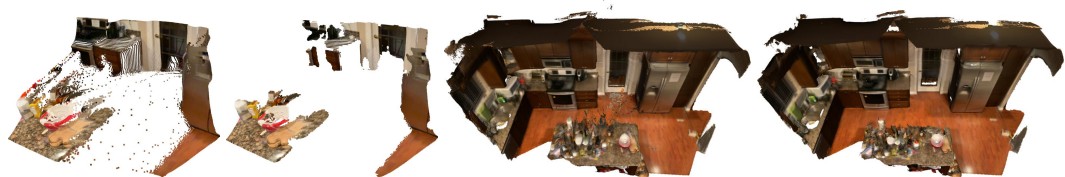

Figure 2: Depth map filtering. From left: raw acquired depth map, filtered depth map, reconstruction result with raw depth maps, reconstruction result with filtered depth maps. Filtering of low-confidence and rapidly changing depth values leads to fewer floating point artifacts.

## A.2   Reconstruction and processing details

A web interface is available for users to browse the scanned data and to initiate the reconstruction process. Users can first preview the geometry of the untextured mesh in an early stage to confirm the quality of reconstruction based on a scan. After the mesh is textured, users can start annotating scenes with our semantic annotation framework.

The reconstruction process consists of a pre-processing step where depth filter is applied to filter out noise and outliers in the depth maps. Then, a dense surface reconstruction is performed using Open3D [20] to obtain 3D mesh geometry which is subsequently decimated, cleaned, and aligned to a global coordinate frame. The cleaned mesh is then textured using and the mesh is textured using the multi-view stereo texturing approach of Waechter et al. [14]. Finally, we apply an unsupervised segmentation based on normals and colors to provide an initial set of coarse and fine segmentations for our annotation interface.

**Depth map filtering.** Depth maps from mobile devices tends to be low resolution and noisy. The raw acquired depth maps contains noise and outliers, especially in edge boundaries with big depth difference, which will introduce artifacts in the reconstruction results as shown in Figure 2. We compute pixel-wise depth differences between pairs of frames to filter out depth values with depth difference greater than 5 cm. In addition, we only use depth pixels with high confidence: confidence of 2 (high) in ARKit for all the data reported in the main paper.

**Reconstruction parameters.** For reconstruction, we use a CUDA-accelerated implementation of volumetric fusion [4] from Open3D [20]. For our dataset, we use the device provided camera poses, and integrate the depth maps during the first pass for reconstruction. In our experiments, we found that a voxel size of 0.01m , and truncation value of 0.08m can produce good results across different scenes. We set the block resolution to 24, and block count to 30000.

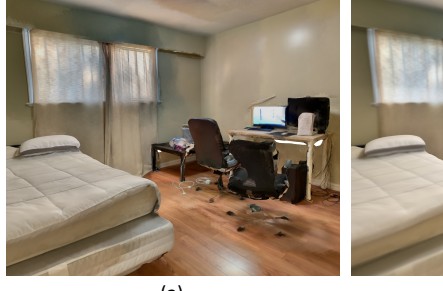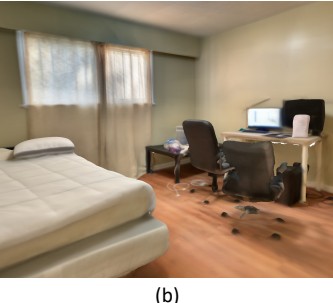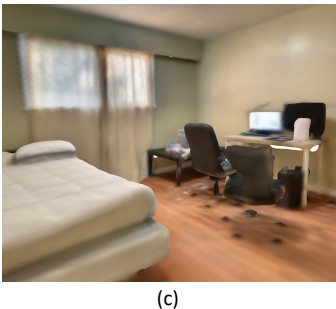

(a)         (b)         (c)

Figure 3: Comparison of reconstruction at different stages: (a) final textured mesh used in annotation pipeline; (b) initial reconstruction with vertex colors (used for quick visualization); and (c) simplified initial mesh (used as input to texturing step). Note that high-frequency color detail is preserved in the final textured mesh relative to the more lightweight initial reconstruction with vertex colors.

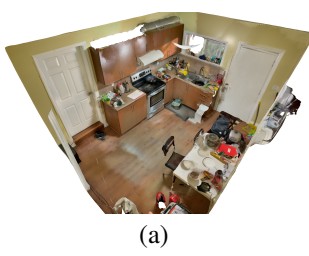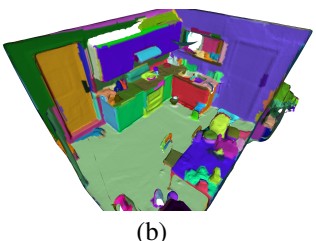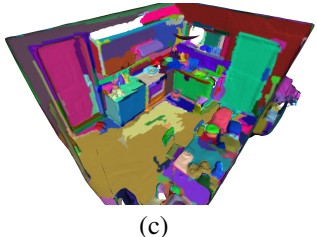

(a)         (b)         (c)

Figure 4: Comparison of our graph-based unsupervised segmentation method with the method used by ScanNet [5]. From left: (a) input textured mesh; (b) segmentation using only vertex normal differences (i.e. method used by ScanNet); and (c) our two-stage vertex normal and color segmentation. Note that geometrically similar but visually distinct objects such as the closed door aligned with the wall in the top right are segmented correctly with our approach.

**Mesh decimation.** We use Instant meshes [7] to simplify the reconstructed meshes. Instant meshes produces simplified triangle meshes by optimizing both edge orientations and vertex positions, and giving high isotropy while preserving sharp features. The simplified mesh helps to reduce the computing complexity of post-processing steps. We use the default settings from Instant meshes, i.e. the simplified mesh has around 1/16 number of vertices of the input mesh.

**Mesh cleaning.** We then use MeshLab [3] to remove duplicated faces, faces with zero area, non-manifold edges, and isolated pieces with small number of faces ($< 50$).

**Mesh texturing.** The earlier reconstruction stages only produce vertex colors for the triangle mesh extracted from marching cubes by averaging color values falling in the same voxel. As a result, the colors are blurry and do not capture high-frequency detail well (e.g., wood floor). In the mesh texturing step we use the high-resolution RGB frames and camera poses to generate textures for the reconstructed geometry. We use the MVS texturing approach of Waechter et al. [14] on every 10th RGB frame from the captured sequence. We empirically set relevant parameters to give good results for the scans we obtained: 'area' based data term, Gauss clamping for photometric outlier removal, maximum and minimum texture atlas dimensions of 8192 and 256 respectively, padding by 7 texels, and waste ratio of 1.0. We downscale texture atlases to 2048x2048 resolutions for the purpose of data loading efficiency. The output of this step is a high quality textured mesh that preserves high frequency color detail (see Figure 3).

**Unsupervised pre-segmentation.** As input to our semantic segmentation interface we use a unsupervised method to extract triangle clusters into a two-tier segment hierarchy. We adapt and extend the approach used by ScanNet [5]. The original method only uses vertex normals to weight the vertices. One limitation is it is hard to separate objects with similar normals, e.g., closed doors and the wall. So we add vertex color properties to the weights, and apply a two-stage segmentation method. We convert texture color in the textured meshes to vertex colors using MeshLab, then construct an undirected graph $G = (V, E)$, where the vertices $v_i \in V$ are the mesh vertices and $e_i \in E$ are the

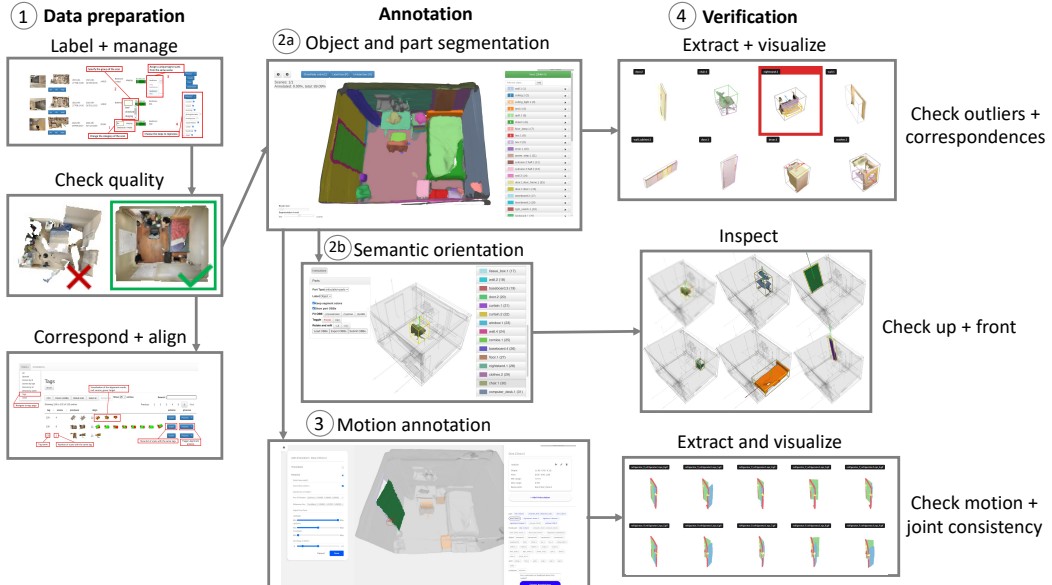

Figure 5: Our annotation framework consists of the following phases: 1) data preparation, 2) semantic annotation of objects and parts, 3) motion annotation, and 4) verification. For 1), we create a management UI for labeling scans, and inspecting reconstructed scan quality. Scans that pass the reconstruction quality check are then corresponded and aligned, and continue to the annotation phases. During annotation, the scan is semantically segmented and objects and parts are labeled (2a). The objects are then semantically annotated with their up and front directions (2b). Objects and parts are also annotated with their motion parameters (3). At the end of each annotation phase, verification (4) is performed, and errors are corrected if necessary.

mesh edges. The weight of the edges is the dot product of the vertex normals $w_{n_{i,j}} = dot(n_i, n_j)$, or color difference $w_{c_{i,j}}$ calculated following Toscana et al. [12]. We first segment the mesh with weights from vertex normals. Then for each segmented cluster, we apply a second segmentation step with edge weights from vertex colors. With this two-stage segmentation, we are able to separate surfaces with both geometric and color differences. Figure 4 shows the effect of this two-stage segmentation. With vertex colors being added into the weights, we can segment objects with similar normals but different surface textures, e.g., the right door on the wall in (c).

## A.3 Annotation details

Our annotation framework consists of several phases: 1) reconstruction quality check and scan alignment; 2) semantic annotation of objects and parts; 3) motion annotation; and 4) verification. During semantic annotation, we densely annotate objects in the scene with object and part labels. We also annotate a semantic front and up direction for each object to produce semantically-oriented bounding boxes (OBBs) that give the pose of the object in each scene. Figure 5 illustrates the annotation flow. We implement our system using a combination of Open3D [20] for alignment, Vue.js[1] for scan management web interface, and three.js[2] and WebGL for the interactive 3D annotation tools.

**Reconstruction quality check and scan alignment.** We conduct a manual quality check on each reconstruction. Verifiers check that the reconstruction is relatively complete with reasonably good geometry (no large holes, significant floating geometry, or misalignments) and with good texturing. Scans that pass the quality check are marked for annotation, and assigned a scene type. All scans of the same scene are manually tagged with the same scene ID. The coordinate frames of the scans are aligned using a semi-automatic approach. A reference scan is first selected for alignment, with the other scans automatically aligned to the reference scan using the multiscale ICP algorithm [1]. The

---

[1]https://vuejs.org/
[2]https://github.com/mrdoob/three.js

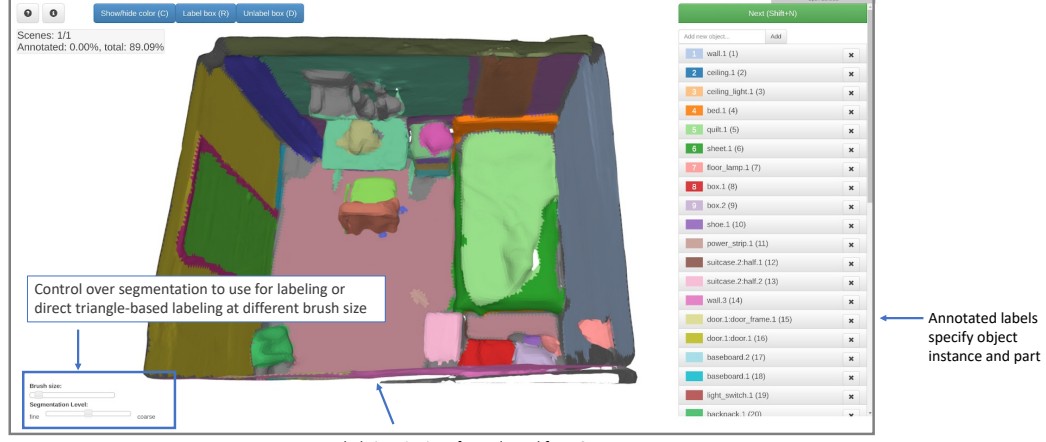

Figure 6: We develop an annotation tool adapted from ScanNet [5] for specifying semantic annotation at the part level by providing a hierarchical fine-grained triangle-based labeling of textured meshes.

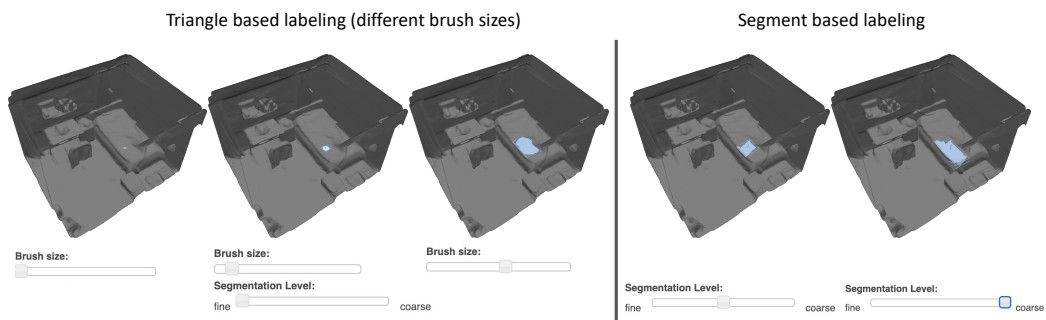

Figure 7: Annotators label parts of the scan using either pre-segmented clusters of triangles or by using triangle-based labeling at different brush sizes to label patches of connected triangles within a given radius.

alignment is then checked and potentially refined by manually selecting corresponding points from the two scans to estimate an alignment with the Umeyama algorithm [13]. This alignment between scans can be leveraged during semantic annotation to propagate annotations from the reference scan to another scan. Of the 369 scans we captured, we kept 273 for our dataset and filtered out 96 scans due to low quality geometry or textures.

**Object and part instances.** To reduce the effort of manual annotation we design a hierarchical labeling interface that allows the user to paint at the triangle level or use the automatically computed segmentations. We develop our interface by adapting the ScanNet [5] segment annotation tool so that labels are accumulated on triangles and the hierarchical labeling is supported (see Figure 6).

For painting at the triangle level, the user can select brushes of different sizes that will label patches of connected triangles within a given radius. The segmentations are at different levels of granularity (normal-only segmentation vs normal and color-based segmentation) to make it easier for the user to select larger regions. Figure 7 illustrates the different granularities at which the annotator can label: using triangle-based brushes or pre-computed segmentations. Annotators are instructed to specify part level annotations for objects that can be articulated. Annotators provide a label of the form `object_id:part_id = object_category.object_index:part_category.part_index` that is used to identify the object and part category and instance. Corresponding objects and parts between scans are specified with the same `object_id` and `part_id`. During annotation, fine-grained object categories are used which are later grouped into coarser categories for our tasks.

For each scene, we also annotate semantically aligned oriented bounding boxes (OBBs) for each object (see Figure 8). Annotators view automatically suggested up (aligned to scene up axis) and front

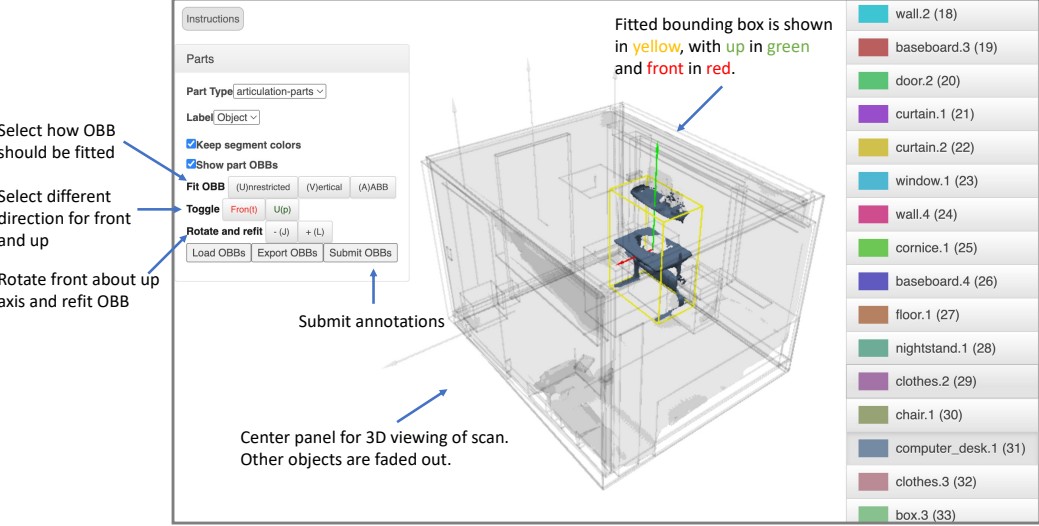

Figure 8: We develop an interface for annotating the semantic up and front of each object. An semantically oriented bounding box (OBB), is automatically fitted to the object based on the specified up and front.

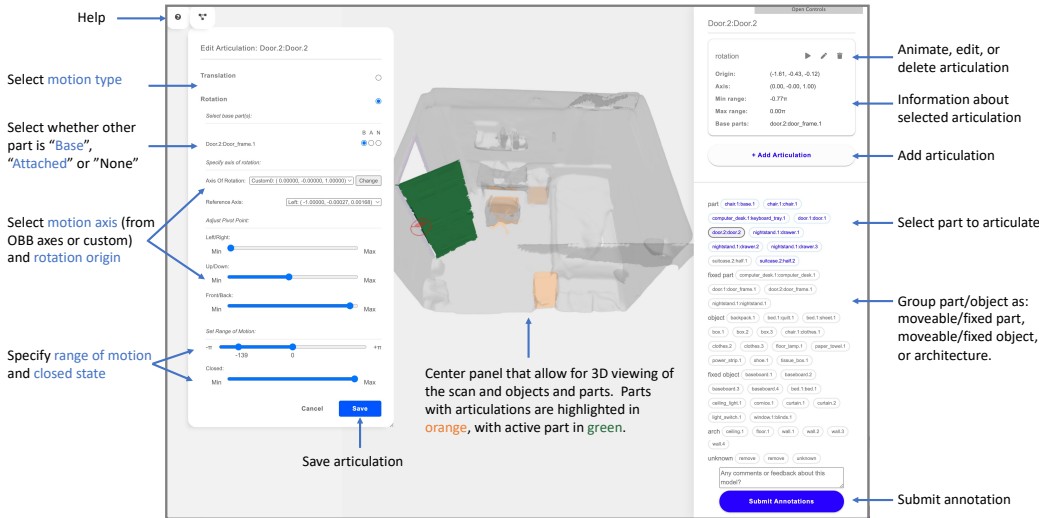

Figure 9: We develop an interface for annotating part articulations based on Xu et al. [18]. Our annotation interface allows annotators to specify the motion type, axis, range, and the closed state. The specification of the closed state allows us to provide semantically meaningful motion state after postprocessing. It also allows for the editing of the connectivity graph by allowing the annotator to specify whether a different part is a 'base', 'attached' or 'none'. Parts and objects are also organized into groups of (moveable) or fixed parts, (moveable) or fixed objects, and architecture elements.

vectors based on an OBB fit for each object, and manually adjust these two directions. The OBBs are automatically re-fitted to the instance annotation based on the annotated up and front vectors.

**Motion parameters.** We develop an interface for annotating part articulations based on Xu et al. [18]'s approach (see Figure 9). Using the part instance annotations, we segment each object mesh into parts to create a part connectivity graph. The connectivity graph indicates the kinematic chain of the articulated parts, and can be adjusted during annotation. The annotator then indicates movable parts, fixed parts, movable objects, fixed objects, and architecture elements (e.g., walls). For each movable part, the annotator specifies articulation parameters: the base part (selected from the connected parts),

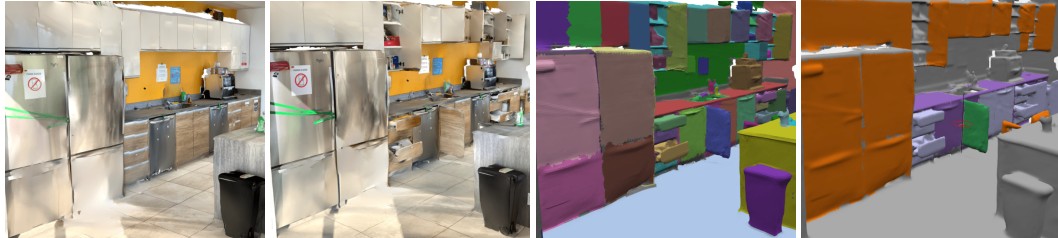

Figure 10: Left and middle left: two scans of a kitchen in different states. Middle right: part-level semantic instance annotation of objects and their parts. Right: articulated parts highlighted in color.

Table 1: Comparison of MultiScan dataset to prior work scene datasets that contain multiple object states. MultiScan provides more scans of more real-world scenes, totaling a significantly larger floor area and navigable area in meters squared. The navigation complexity and scene clutter metrics as defined by Ramakrishnan et al. [11] are also high, indicating many large furniture-sized objects and many smaller table-top objects, respectively. Finally, MultiScan provides significantly more corresponded object observations of objects in different states (reported in the Object corrs. column), and is the only dataset that also provides corresponded observations of object parts in different states (Part corrs. column).

| Dataset | Scenes | Scans | Floor area | Nav. area | Nav. complexity | Scene clutter | Object corrs. | Part corrs. |
|---|---|---|---|---|---|---|---|---|
| Rescan [6] | 13 | 45 | 668.9 | 169.7 | 2.33 | 2.84 | 691 | ✗ |
| RIO10 [15] | 10 | 74 | 1955.7 | 503.2 | **4.41** | 3.82 | 2739 | ✗ |
| MultiScan | **117** | **273** | **8566.1** | **3177.3** | 4.20 | **4.15** | **9977** | **4667** |

motion type (translation or rotation), axis of motion, motion origin (for rotation), and motion range. For consistent and meaningful motion states, we define positive rotation to be counter-clockwise about the motion axis, and positive translation to be in the direction of the motion axis. We also define the closed state to have motion value of 0. Prior datasets either lack specification of the motion range [8, 16] or do not have a semantically meaningful motion range [17, 19]. The annotation interface allows the annotator to check the movement of the articulated part by viewing the animation of the part in motion.

**Verification.** After each phase of annotation is complete, we conduct a verification pass to identify and fix annotation issues. For the segmentation and labeling of object and parts, we run the `pyenchant`[3] spelling checker followed by manual correction to ensure category labels are correctly spelled. We also take the OBBs and compute statistics over the grouped labels to compute the average and standard deviation of the OBB volume. We manually check outlier objects (volume is outside 2 standard deviations of the mean). We also check floors, walls, and ceilings to ensure that that are mostly vertical or horizontal surfaces. Finally, we extract and render individual objects, as well as their parts in different states, and visually inspect them.

## B   Additional dataset statistics

Figure 10 shows first-person views of two scans of a scene and associated annotations for the objects and their parts. Figure 11 provides several more examples visualized from external viewpoints to show the entirety of each scene. To quantify the value of the MultiScan dataset relative to datasets from prior work that include multiple observations of real scenes, we compute a number of statistics that summarize the scale, structural complexity, and volume of corresponded object and part instance observations. To measure scale and structural complexity we use floor and navigable area numbers as well as the navigation complexity and scene clutter metrics reported by Ramakrishnan et al. [11]. See Table 1 for these comparative statistics. The MultiScan dataset has considerably more scenes and scans, resulting in higher total areas. The scenes themselves have higher navigational complexity (indicating many furniture pieces and other large objects), and fairly high scene clutter (indicating many smaller and table-top objects). Moreover, MultiScan has far more corresponded observations

---

[3]https://pyenchant.github.io/pyenchant/

| Textured mesh reconstruction | Semantic instances | Semantic OBBs | Part articulations |
| :---: | :---: | :---: | :---: |

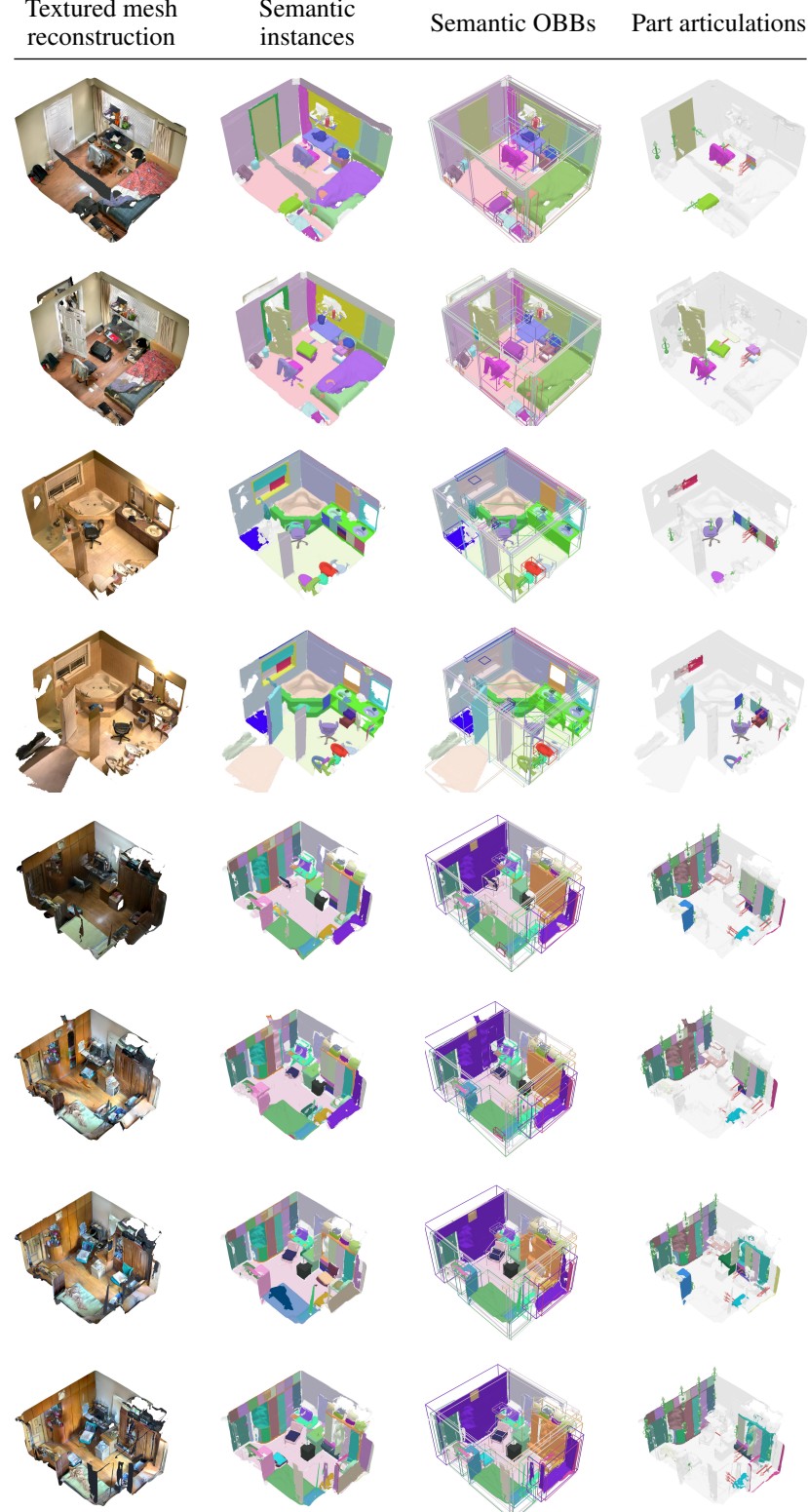

Figure 11: Examples of annotated scans from the MultiScan dataset. Each scan is reconstructed as a textured mesh, which is then annotated with object and part semantic instances, semantic oriented bounding boxes (OBBs), and part articulations.

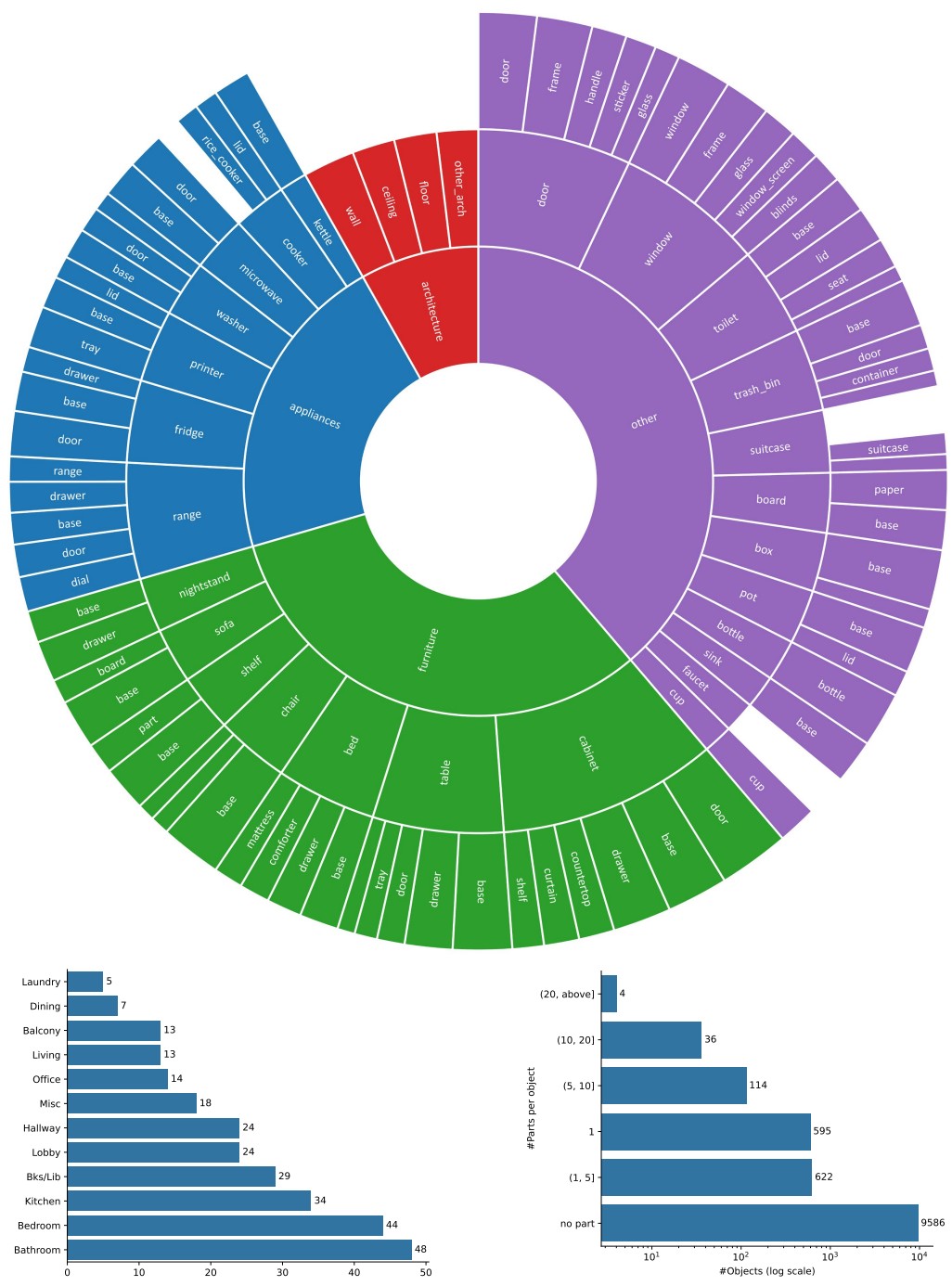

Figure 12: Top: i) distribution over object supercategory (inner ring), object category (middle ring), and part category (outer ring), only object categories with more than 30 instances are shown; ii) Bottom left: distribution of scans over room categories. Bottom right: distribution for number of articulated parts per object (rigid objects are indicated as having no parts).

Table 2: Number of scenes, scans, objects, and parts used across splits for the MultiScan segmentation tasks. The part mobility prediction task uses the same set of objects as the part segmentation task.

| Task | Scenes | | | | Scans | | | | Objects | | | | Parts | | | |
|---|---|---|---|---|---|---|---|---|---|---|---|---|---|---|---|---|
| | Train | Val | Test | **Total** | Train | Val | Test | **Total** | Train | Val | Test | **Total** | Train | Val | Test | **Total** |
| Obj. Seg. | 61 | 20 | 20 | 101 | 174 | 42 | 41 | 257 | 2234 | 411 | 389 | 3034 | – | – | – | – |
| Part Seg. | 44 | 19 | 18 | 81 | 140 | 40 | 37 | 217 | 666 | 148 | 131 | 945 | 2447 | 589 | 460 | 3496 |

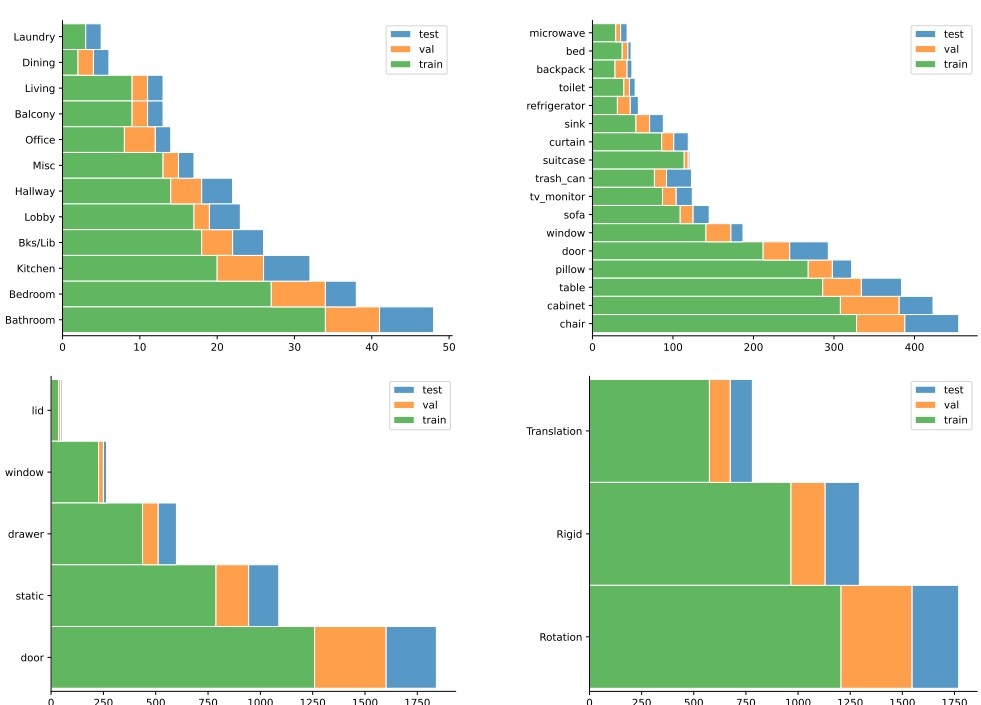

Figure 13: Plots providing the distribution over room types (top left), over object categories (top right), over part categories (bottom left), and over articulation motion types (bottom right). The colors indicate train, val and test split membership. All these statistics are for the segmentation task datasets we use for our experiments.

of objects, and is unique in that it has corresponded observations of object parts in different states as well.

As our interface allows for more fine-grained labeling than what is possible purely with the pre-computed segmentations, we check how many of our annotations leverage fine-grained labeling. We find that over 90% of annotations (93% ignoring noisy reconstruction patches marked for removal) required cutting of the coarse segmentation, and 83% (88% ignoring removals) required cutting of the finer segmentation utilizing color. At the segment level, 34% of coarse segments were cut and 28% of the finer segments were cut. This shows that the segmentation was helpful in providing larger patches that can be labeled as a unit, but that more precise cutting is required to get more accurate object and part boundaries.

In Figure 12 we plot distributions over object categories and part categories, as well as distributions over scene categories and the number of parts per object instance. The object and part category distributions in particular show a breadth of objects and associated parts acquired in the MultiScan dataset. A more detailed breakdown showing the distributions over scene categories, object categories, part categories, and articulation types is provided in Figure 13.

Table 3: Object instance segmentation results on MultiScan val set. Values are the $AP$, $AP_{50}$ and $AP_{25}$ metrics (higher is better). We also report the standard error.

| metric | method | all | door | table | chair | cab | win | sofa | pillow | tv | curt | bin | sink | bpk | bed | fridge | toilet |
|---|---|---|---|---|---|---|---|---|---|---|---|---|---|---|---|---|---|
| $AP$ | PG [9] | 26.2±0.7 | **35.9**±1.1 | 22.0±3.2 | 65.8±1.9 | 16.0±2.9 | **10.3**±2.8 | 29.2±7.8 | 10.8±1.4 | 24.7±3.9 | **2.6**±1.4 | 32.8±3.5 | 43.9±3.9 | 1.4±0.7 | 60.5±5.0 | **16.8**±4.6 | 64.3±5.6 |
| | SSTNet [10] | **32.6**±1.0 | 30.2±2.0 | **28.5**±1.0 | **69.5**±2.6 | 18.8±1.3 | 8.6±1.2 | **50.6**±4.6 | **18.8**±3.1 | 38.4±3.3 | 0.7±0.4 | **42.9**±8.4 | 44.5±2.4 | **15.8**±8.0 | **66.4**±2.3 | 11.7±1.7 | **83.1**±4.1 |
| | HAIS [2] | 30.1±0.6 | 32.3±1.1 | 27.9±1.9 | 67.3±1.0 | **20.6**±0.3 | 7.5±1.6 | 37.1±2.9 | 13.4±3.7 | **40.7**±4.3 | 0.4±0.3 | 32.5±7.8 | **52.7**±6.1 | 8.1±3.9 | 59.4±3.2 | 13.9±2.4 | 72.9±7.2 |
| $AP_{50}$ | PG [9] | 43.3±1.1 | **68.0**±3.2 | 32.3±3.1 | **83.1**±1.0 | 32.1±3.9 | **26.9**±7.9 | 56.9±17.9 | 27.7±6.9 | 43.2±3.9 | **7.1**±2.9 | 50.4±6.2 | 82.8±3.0 | 4.9±1.9 | **100.0**±0.0 | **35.4**±7.5 | 76.2±4.8 |
| | SSTNet [10] | 46.0±1.8 | 51.8±0.5 | **38.7**±2.4 | 81.6±2.4 | 34.1±1.5 | 22.9±5.5 | 70.5±8.7 | 34.4±5.6 | 52.9±3.4 | 3.9±2.2 | 53.8±11.4 | 72.5±6.1 | **23.3**±11.8 | 99.4±0.6 | 15.9±1.9 | **90.5**±4.8 |
| | HAIS [2] | **49.2**±0.8 | 59.1±2.5 | 38.3±2.4 | 82.7±2.0 | **38.2**±1.7 | 19.8±3.8 | **73.2**±5.3 | **36.2**±6.8 | **58.8**±5.9 | 1.6±0.7 | **57.9**±9.3 | **88.1**±0.1 | 21.6±6.6 | 98.6±1.4 | 32.7±5.7 | 88.9±5.7 |
| $AP_{25}$ | PG [9] | 54.7±1.6 | **83.6**±2.3 | 50.8±3.8 | **87.1**±1.2 | 58.1±2.3 | 50.5±5.4 | 78.8±8.5 | 33.3±5.5 | 50.8±3.7 | **24.3**±8.6 | 57.5±7.4 | **88.5**±2.4 | 18.5±11.0 | **100.0**±0.0 | 42.5±6.3 | 85.1±7.7 |
| | SSTNet [10] | 55.0±1.2 | 71.8±2.3 | 54.0±2.6 | 86.6±1.6 | 53.3±0.6 | 43.1±7.3 | **87.0**±0.3 | **43.7**±4.5 | 54.9±5.2 | 15.3±4.0 | 55.3±10.2 | 84.7±1.8 | 26.1±13.0 | 99.4±0.6 | 17.0±1.1 | **99.4**±0.6 |
| | HAIS [2] | **57.9**±0.6 | 79.1±1.0 | **54.7**±3.6 | 84.4±1.9 | **61.5**±1.9 | 43.4±5.0 | 83.2±4.1 | 39.7±6.2 | **64.5**±0.2 | 14.5±4.7 | **63.9**±15.1 | 88.1±0.1 | **32.4**±10.8 | 98.6±1.4 | **43.5**±3.4 | 92.5±4.1 |

Table 4: Object instance segmentation results on MultiScan test set. Values are the $AP$, $AP_{50}$ and $AP_{25}$ metrics (higher is better). The standard error is also reported.

| metric | method | all | door | table | chair | cab | win | sofa | pillow | tv | curt | bin | sink | bpk | bed | fridge | toilet |
|---|---|---|---|---|---|---|---|---|---|---|---|---|---|---|---|---|---|
| $AP$ | PG [9] | 20.8±4.7 | **34.8**±4.0 | 27.6±1.6 | 68.8±1.7 | 8.3±2.8 | **5.1**±2.89 | 1.7±0.98 | 17.1±2.88 | 25.4±3.23 | 3.8±1.31 | 19.5±2.53 | 29.0±5.21 | 0.0±0.0 | 33.0±6.45 | 3.1±2.54 | 76.0±9.3 |
| | SSTNet [10] | **27.7**±0.4 | 25.6±1.5 | **32.0**±2.3 | **80.5**±1.5 | 9.4±2.1 | 2.8±0.4 | **9.3**±1.2 | **29.4**±3.6 | **59.4**±3.8 | **6.8**±2.9 | 19.1±5.9 | **38.5**±5.0 | 0.0±0.0 | **55.2**±6.1 | **8.6**±6.0 | **93.7**±3.0 |
| | HAIS [2] | 22.9±0.8 | 33.0±2.3 | 31.3±1.3 | 72.2±0.5 | **12.7**±0.3 | 3.0±0.9 | 1.4±0.8 | 17.3±0.5 | 43.5±2.5 | 4.9±2.1 | **21.2**±2.0 | 34.5±1.7 | 0.0±0.0 | 30.8±5.5 | 2.9±2.5 | 79.6±6.7 |
| $AP_{50}$ | PG [9] | 35.8±1.5 | 53.9±3.7 | 41.1±2.7 | 81.6±0.9 | 20.3±4.1 | **11.3**±4.2 | 5.4±2.0 | 42.2±5.2 | 45.1±7.9 | 6.6±1.0 | 33.5±5.9 | 71.4±6.4 | 0.0±0.0 | 91.7±8.3 | 8.6±4.6 | 95.2±4.8 |
| | SSTNet [10] | **41.9**±0.7 | 50.3±1.5 | 43.6±2.6 | **85.3**±1.6 | 19.9±4.6 | 9.0±1.4 | **30.1**±1.3 | **54.4**±5.0 | **77.8**±3.5 | **13.9**±4.6 | 30.7±4.2 | **80.8**±8.1 | 0.0±0.0 | **98.1**±1.9 | **15.7**±8.1 | **99.4**±0.6 |
| | HAIS [2] | 35.3±1.0 | **57.8**±0.1 | **45.0**±1.4 | 80.1±2.1 | **25.5**±1.5 | 11.6±0.9 | 7.8±3.8 | 35.1±3.5 | 62.2±3.6 | 6.8±2.2 | **33.7**±3.2 | 73.3±5.1 | 0.0±0.0 | 58.3±8.3 | 7.5±6.3 | 95.2±4.8 |
| $AP_{25}$ | PG [9] | 46.9±0.3 | 72.4±2.9 | **56.9**±2.1 | 83.9±1.7 | 40.8±1.5 | 31.6±8.6 | 29.6±1.6 | 58.9±1.2 | 56.5±6.8 | 9.5±2.4 | **39.6**±6.5 | 87.6±3.6 | 0.0±0.0 | **100.0**±0.0 | **26.9**±3.5 | **100.0**±0.0 |
| | SSTNet [10] | **50.2**±1.3 | 69.1±3.8 | 56.2±3.3 | **87.2**±1.5 | 39.2±3.7 | **26.8**±1.7 | **47.6**±3.8 | **64.2**±5.5 | **81.6**±4.5 | **19.5**±4.5 | 34.9±3.4 | 89.0±2.6 | **18.1**±8.4 | 98.1±1.9 | 20.0±11.6 | 99.4±0.6 |
| | HAIS [2] | 47.1±1.2 | **73.0**±1.4 | 60.4±1.0 | **87.2**±2.0 | **45.8**±2.7 | 24.7±3.5 | 30.9±1.7 | 44.0±0.8 | 78.4±5.8 | 9.2±1.9 | 39.5±4.8 | **90.2**±2.0 | 0.0±0.0 | **100.0**±0.0 | 16.2±10.9 | 95.2±4.8 |

# C  Additional experiments and results

In the main paper we reported results on the validation set. In this supplement we provide more details on the data split, additional ablations on the validation set, and experimental results on the test set.

**Data splits.** The split by scene allows us to evaluate the consistency of instance segmentation across scans, and to ensure that the val and test splits do not have scenes from the training split. See Table 2 for the number of scenes, scans, labeled objects and parts across splits and Figure 13 for distribution over categories. Note that the part segmentation task discards scans with zero parts in the part-level semantic label set.

## C.1  Object and part instance segmentation

For object and part instance segmentation, we report the mean Average Precisions (mAP) at different thresholds of IoU for each category and averaged across the categories. $AP_{25}$ and $AP_{50}$ denote the AP scores at IoU thresholds of 0.25 and 0.5, with $AP$ denoting the average score with IoU thresholds set from 0.5 to 0.95. In the main paper we reported mean $AP$ results for the validation set over runs with three different random seeds. Here, we report the mean and standard error of $AP$, $AP_{50}$, and $AP_{25}$ for both the validation and test sets.

**Object segmentation.** Object instance segmentation results on the validation set are in Table 3 and on the test set are in Table 4. As expected, $AP_{25}$ and $AP_{50}$ performance is higher than overall $AP$, with performance of chair, sink, bed, and toilet being especially high. We note that on the validation set, HAIS has higher $AP_{25}$ and $AP_{50}$ than SSTNet but lower overall $AP$ when we sweep the IOU threshold from 0.5 to 0.95, indicating that HAIS has lower performance at higher IOUs.

Table 5: Part instance segmentation results on the MultiScan val set. Values are the $AP$, $AP_{50}$ and $AP_{25}$ metrics (higher is better). We also report the standard error on the mean $AP$ values across all categories. Results reported using ground-truth (left) and predicted (right) object instance segmentations. When using predicted objects, we use the same method for both object and part level segmentation.

| metric | method | GT segmentation | | | | | | predicted segmentation | | | | | |
|---|---|---|---|---|---|---|---|---|---|---|---|---|---|
| | | all | static | door | drwr | win | lid | all | static | door | drwr | win | lid |
| $AP$ | PG [9] | 24.8±0.5 | 56.5±1.2 | 26.5±0.7 | 4.8±1.1 | **0.1**±0.0 | 36.0±1.7 | 8.2±0.3 | **9.7**±0.2 | 8.8±0.1 | 0.3±0.2 | 0.0±0.0 | 22.2±1.4 |
| | SSTNet [10] | **29.8**±0.1 | 53.4±1.7 | **35.0**±0.6 | **12.0**±0.8 | **0.1**±0.1 | **48.4**±1.0 | **9.5**±0.5 | 8.5±0.3 | 6.6±0.3 | 0.8±0.3 | 0.0±0.0 | **31.6**±2.5 |
| | HAIS [2] | 24.6±1.0 | **58.2**±2.4 | 23.3±0.9 | 8.1±0.6 | 0.0±0.0 | 33.4±5.4 | 9.1±0.6 | 8.3±0.4 | **9.4**±0.6 | **1.8**±0.3 | 0.0±0.0 | 26.3±2.7 |
| $AP_{50}$ | PG [9] | 42.3±1.0 | 89.9±1.6 | 44.1±1.4 | 11.7±1.9 | **0.7**±0.2 | 65.1±7.3 | 16.6±0.2 | **22.8**±0.2 | 16.3±0.3 | 1.0±0.4 | 0.0±0.0 | 42.7±0.8 |
| | SSTNet [10] | **46.4**±0.7 | 82.0±1.5 | **51.4**±0.8 | **24.4**±0.6 | 0.2±0.2 | **74.1**±3.7 | 15.5±0.3 | 21.1±0.5 | 11.4±0.3 | 1.4±0.7 | 0.0±0.0 | 43.6±0.8 |
| | HAIS [2] | 40.3±1.4 | **90.6**±2.0 | 38.7±1.5 | 13.9±0.7 | 0.3±0.3 | 58.0±6.9 | **18.1**±0.5 | 19.0±1.0 | **17.4**±0.6 | **2.7**±0.0 | 0.0±0.0 | **51.6**±1.9 |
| $AP_{25}$ | PG [9] | **52.5**±0.9 | **99.0**±1.0 | 58.1±0.8 | 18.9±1.3 | **8.7**±2.2 | 77.8±0.0 | 25.8±0.3 | **42.4**±0.5 | 23.3±0.3 | **5.6**±0.8 | **2.5**±1.4 | 55.6±0.0 |
| | SSTNet [10] | 51.9±0.3 | 95.0±0.3 | **58.9**±1.2 | **27.0**±1.2 | 0.9±0.4 | 77.8±0.0 | 22.1±0.3 | 36.2±0.4 | 16.4±0.9 | 2.2±0.5 | 0.0±0.0 | 55.6±0.0 |
| | HAIS [2] | 49.3±0.5 | 98.9±0.3 | 46.9±1.4 | 18.4±0.5 | 4.6±1.9 | 77.8±0.0 | **26.4**±0.2 | 37.3±1.0 | **21.5**±0.4 | 4.2±0.6 | 2.2±1.7 | **66.7**±0.0 |

| Input | GT | PG [9] | SSTNet [10] | HAIS [2] |

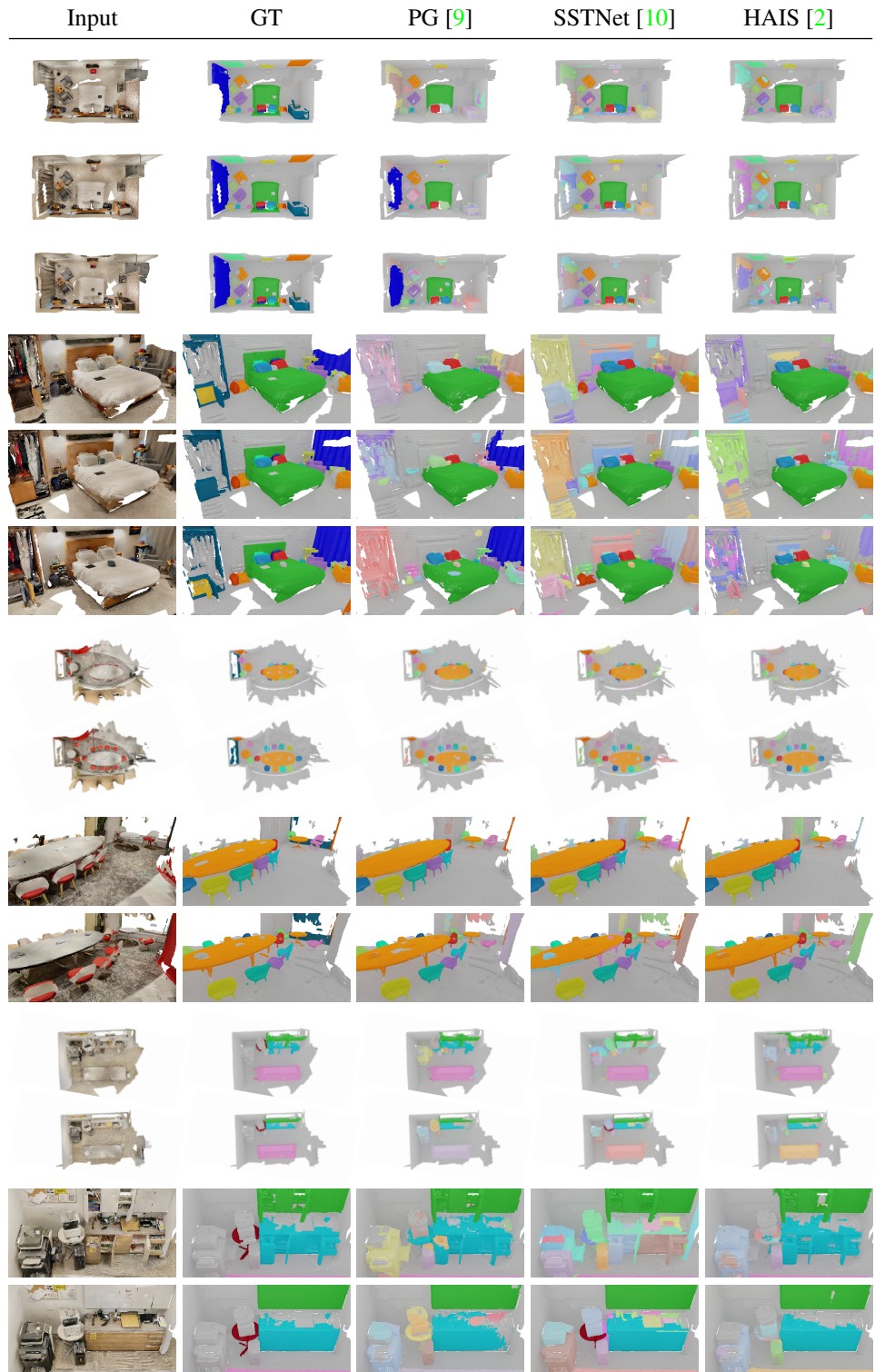

Figure 14: Additional examples of object instance segmentation results from the MultiScan val set. Colors match ground truth when instance categories are correctly predicted, and are different from the ground truth otherwise.

| Input | GT | PG [9] | SSTNet [10] | HAIS [2] |
|-------|----|----|----|----|

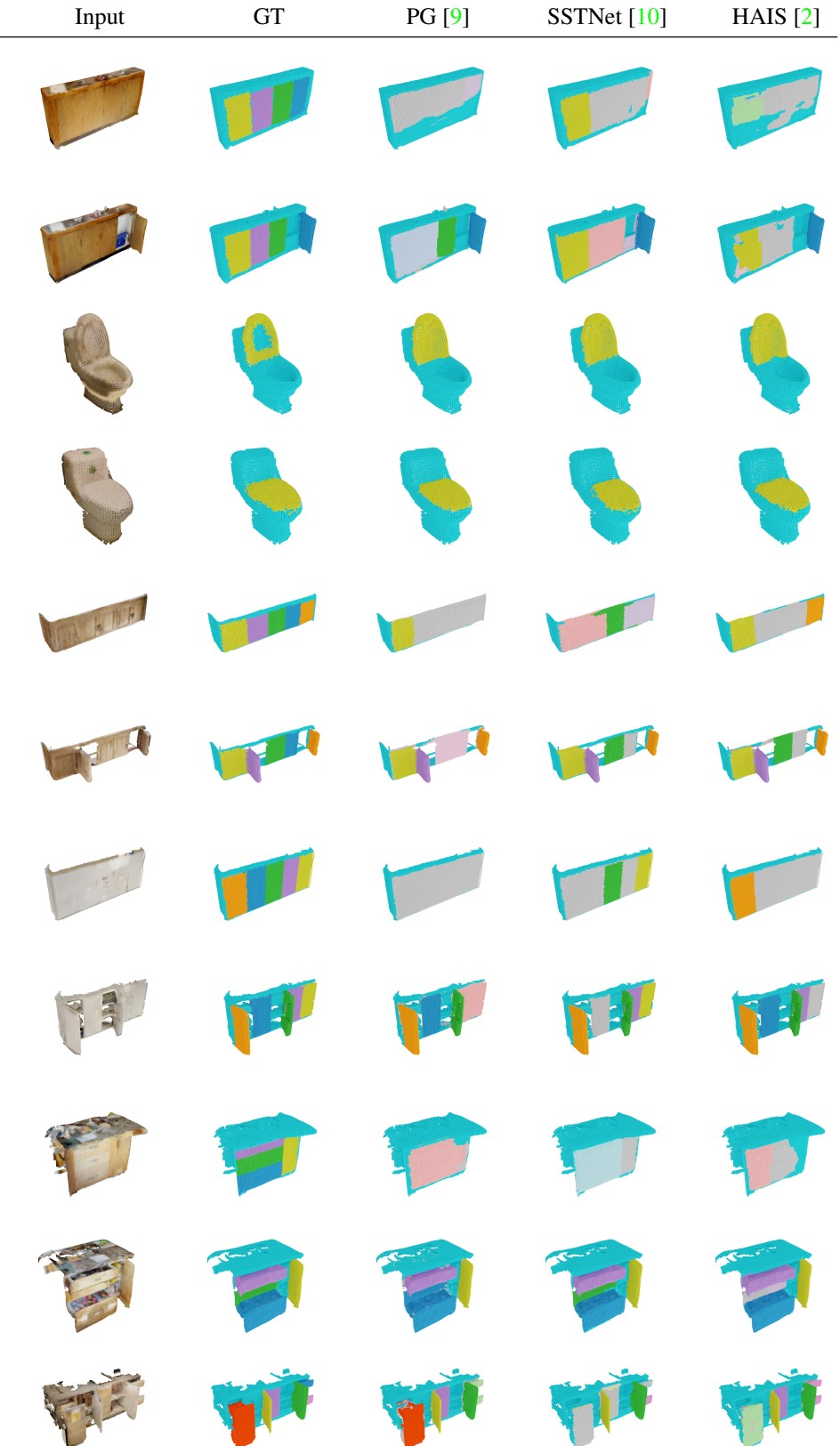

Figure 15: Additional examples of part instance segmentation results from the MultiScan val set. Colors indicate different parts. Correctly predicted parts match the ground truth colors, and are different otherwise.

Table 6: Part instance segmentation results on the MultiScan test set. Values are the $AP$, $AP_{50}$ and $AP_{25}$ metrics (higher is better). Results reported using ground-truth (left) and predicted (right) object instance segmentations. When using predicted objects, we use the same method for both object and part level segmentation.

| metric | method | GT segmentation | | | | | | predicted segmentation | | | | | |
|---|---|---|---|---|---|---|---|---|---|---|---|---|---|
| | | all | static | door | drwr | win | lid | all | static | door | drwr | win | lid |
| $AP$ | PG [9] | 30.1±0.8 | 55.4±1.7 | 29.4±0.6 | 3.9±0.6 | 9.3±0.4 | 52.7±2.6 | 9.8±0.4 | 7.2±0.3 | **15.3**±0.6 | 0.2±0.1 | 0.0±0.0 | 26.4±2.3 |
| | SSTNet [10] | 32.0±1.3 | 57.8±0.4 | 38.6±1.2 | 3.5±1.1 | 3.0±2.9 | 57.0±5.1 | 14.3±0.7 | 6.4±0.5 | 8.9±0.6 | 0.0±0.0 | 4.5±1.8 | 51.8±3.4 |
| | HAIS [2] | **33.2**±0.4 | **59.6**±0.5 | **31.6**±1.1 | **5.2**±0.2 | **10.3**±1.1 | **59.0**±2.4 | **17.5**±0.8 | **9.2**±0.1 | 13.4±0.7 | **0.5**±0.3 | **5.9**±2.2 | **58.4**±0.9 |
| $AP_{50}$ | PG [9] | 50.6±0.2 | **84.1**±1.5 | 45.9±0.6 | **9.6**±0.7 | 13.2±0.7 | 100.0±0.0 | 19.2±0.3 | 17.4±0.6 | **23.2**±1.1 | **1.7**±0.4 | 0.0±0.0 | 53.6±2.1 |
| | SSTNet [10] | 49.6±1.2 | 83.2±1.3 | 50.7±2.4 | 6.5±2.2 | 7.5±6.1 | 100.0±0.0 | 27.6±2.3 | 15.1±1.2 | 14.2±0.4 | 0.0±0.0 | 13.5±6.3 | 95.2±4.8 |
| | HAIS [2] | **50.8**±0.6 | 83.0±0.3 | **46.2**±1.8 | 7.9±0.5 | **17.0**±2.3 | 100.0±0.0 | **31.9**±1.7 | **20.8**±0.2 | 21.3±1.4 | 1.5±0.6 | **15.6**±6.9 | **100.0**±0.0 |
| $AP_{25}$ | PG [9] | 57.4±1.0 | **95.4**±0.6 | 56.9±1.3 | 14.2±0.9 | 20.5±4.3 | 100.0±0.0 | 30.5±0.5 | **38.6**±0.4 | **29.4**±1.4 | 4.7±0.2 | 4.5±1.9 | 75.4±4.0 |
| | SSTNet [10] | 58.1±1.0 | 91.6±1.7 | **60.7**±1.9 | 10.8±1.4 | **27.3**±5.3 | 100.0±0.0 | 34.7±1.6 | 28.9±0.7 | 19.9±0.5 | 0.4±0.4 | **29.2**±4.2 | 95.2±4.8 |
| | HAIS [2] | **58.2**±0.5 | 94.2±1.5 | 54.9±2.3 | **15.2**±1.2 | 26.7±3.5 | 100.0±0.0 | **37.1**±0.5 | 33.2±0.6 | 26.0±1.9 | **4.8**±0.5 | 21.5±1.7 | **100.0**±0.0 |

Table 7: Complete object instance segmentation consistency results on val set, including standard error on reported mean values. HAIS provides the most consistent object segmentations across scans of the same scene. Bathroom and office scenes are overall easiest, while bedroom and kitchen scenes are quite challenging likely due to the prevalence of hard-to-segment cabinetry.

| | all | balc | bathroom | bed | bks/lib | dining | hall | kitchen | living | lobby | misc | office |
|---|---|---|---|---|---|---|---|---|---|---|---|---|
| PG [9] | 44.43±4.36 | **29.63**±9.80 | 73.61±4.81 | 30.97±3.82 | 32.94±2.78 | **39.39**±8.02 | **44.44**±7.35 | **39.93**±2.57 | 25.00±0.00 | 63.33±3.33 | **40.74**±3.70 | **68.75**±1.80 |
| SSTNet [10] | 40.91±4.93 | 18.52±7.41 | 62.50±6.36 | **37.45**±3.24 | **40.08**±3.46 | 33.33±8.02 | 41.67±0.00 | 29.24±2.91 | **41.67**±5.51 | 63.33±8.82 | 25.93±3.70 | 56.25±4.77 |
| HAIS [2] | **44.57**±3.52 | 22.22±0.00 | **81.94**±3.67 | 36.16±3.11 | 36.11±1.59 | 36.36±5.25 | 33.33±0.00 | 40.88±4.09 | 33.33±5.51 | **66.67**±3.33 | **40.74**±7.41 | 62.50±4.77 |

This shows the importance of evaluating at higher IoU thresholds as different methods may perform better. We see similar performance trends on the test set as on the validation set. Figure 14 shows additional examples comparing predictions obtained using the different methods.

**Part segmentation.** Part instance segmentation performance is reported on the validation set in Table 5, and on the test set in Table 6. We see that again part segmentation based on predicted object segmentations is incredibly challenging for all methods, with low overall $AP$ scores across the board. Using ground truth segmentations results in higher scores, but there is still a large gap for potential improvements, especially for challenging part categories such as windows, drawers, and doors. Figure 15 shows additional qualitative comparison examples for part instance predictions on the MultiScan val set.

## C.2 Segmentation consistency

In Table 8 we report the object instance segmentation consistency measure we defined in the main paper across scans for the test set (compare to results on the val set from the main paper). The segmentation consistency is also computed over runs with three different random seeds, and we report the standard error in addition to the mean for the val set in Table 7. We again see the overall trend that all approaches struggle to provide consistent segmentations across scans of the same scene.

In addition to these quantitative results, we also provide a variety of qualitative examples visualizing the consistency of segmentations in Figure 16. In this figure, colors matching the ground truth indicate consistently segmented object instances, while incorrect predictions have other colors.

## C.3 Mobility prediction

**Summary of Shape2Motion architecture.** Shape2Motion [16] consists of three stages: 1) mobility proposal, 2) proposal matching, and 3) mobility optimization. The Mobility Proposal Network (MPN, stage 1) proposes a set of masks for the movable parts and a set of motion joints. In stage 1, there is

Table 8: Object instance segmentation consistency on the test set (corresponding to val set results in the main paper). The trends observed here are similar. SSTNet provides the most consistent object segmentations across scans of the same scene.

| | all | balc | bath | bed | bks/lib | din | hall | kitchen | laundry | living | lobby | misc | office |
|---|---|---|---|---|---|---|---|---|---|---|---|---|---|
| PG [9] | 52.11±3.61 | 66.67±6.67 | 55.56±5.23 | 33.17±1.71 | 30.05±2.41 | 87.50±0.00 | 83.33±0.00 | 28.09±3.25 | 66.67±9.62 | 33.33±1.67 | 22.22±2.78 | 66.67±6.42 | 52.11±3.61 |
| SSTNet [10] | **59.06**±3.92 | 66.67±6.67 | **63.92**±6.43 | **34.13**±3.09 | **67.42**±1.16 | **95.83**±4.17 | 83.33±0.00 | **33.86**±2.86 | 61.11±5.56 | **38.33**±4.41 | **34.72**±1.39 | **70.37**±7.41 | **59.06**±3.92 |
| HAIS [2] | 54.42±3.97 | 66.67±6.67 | 56.72±2.84 | 29.60±2.70 | 51.77±4.04 | 83.33±4.17 | 83.33±0.00 | 25.23±1.85 | **83.33**±9.62 | 36.67±1.67 | 15.28±3.67 | 66.67±6.42 | 54.42±3.97 |

| Input | GT | PG [9] | SSTNet [10] | HAIS [2] |
|-------|-----|--------|-------------|----------|

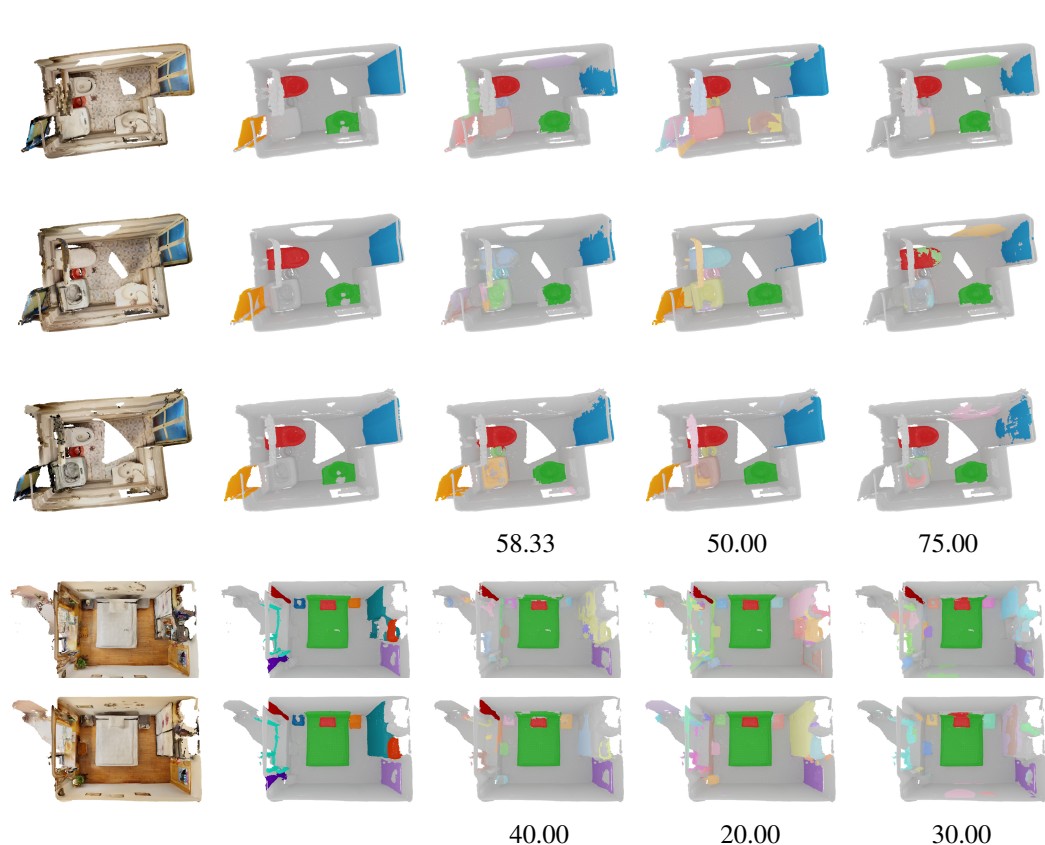

Figure 16: Object instance segmentations for two scenes from the val set. Colors show corresponding instances, indicating prediction consistency across scans of the same scene. Correct predictions (label matches ground truth) have the same color, and incorrect predictions are colored differently. The numbers below each pair of scans segmented by a given method report the overall consistency score for the given method's segmentations of the scans of the same room.

no part-to-joint associations. To find the correct part and joint association, the Proposal Matching Module (PMM, stage 2) assigns scores for the part and joint combinations proposed in stage 1. The scorer for the PMM is trained to score the part and joint combinations by measuring the amount of alignment between the moved part points when simulating movement of the points based on the mobility parameters. The Mobility Optimization Network (MON, stage 3) is used to refine high scoring part segmentation and joint parameter combinations by further simulating the movement of the part. Finally, an extraction step similar to Non-Maximum Suppression (NMS) is used to find non-overlapping motion parts and part parameters to obtain the final set of part segmentations and motion joints.

**Shape2Motion re-implementation.** In the original Shape2Motion architecture's stage 1, the MPN predicts the motion axis and origin by first predicting an anchor and then regressing the residual. The motion origin is regressed from an anchor point that is selected/predicted from the input point cloud. During stage 3, the MON performs a binary segmentation of the movable part of interest and predicts the residual vectors (from the anchor point) for the motion origins and axes. In our re-implementation of Shape2Motion stage 3 (MON) we use the optimized binary mask but discard the predicted residual vectors for the joint origins and axes (we don't add them to the the joint parameters predicted in stage 1), as we found these make the joints prediction results worse.

Our re-implementation of Shape2Motion in PyTorch is significantly faster and easier to run than the original implementation. On the original Shape2Motion dataset, the original implementation training takes 66 hours: 31 hours for MPN (stage 1) and 35 hours for MON (stage 3) for 50 epochs each on

Table 9: We developed a PyTorch re-implementation of Shape2Motion [16]. We compare our re-implementation against the original numbers reported in the paper on the original Shape2Motion dataset. We use a joint score $\geq 0.8$ for determining matched parts for the original Shape2Motion implementation. Also, the original implementation uses position (xyz) and normal (n) features at each point.

| method | IoU↑ | EPE↓ | MD↓ | OE↓ | TA↑ |
|---|---|---|---|---|---|
| Original Shape2Motion [16] | **84.70** | **0.025** | **0.010** | 6.875 | 98.00 |
| Shape2Motion re-implementation (ours) | 80.07 | 0.062 | 0.030 | **1.604** | **99.70** |

Table 10: Ablations using different combinations of per-point features for the input point clouds given to the Shape2Motion architecture. Results on the MultiScan val set, using ground truth object instance segmentations. We evaluate whether incorporating color (+rgb), and normal (+n) features, and using data augmentation (aug) help to improve mobility prediction. We report both the mean and standard error across three random seeds.

| Input | IoU↑ | EPE↓ | MD↓ | OE↓ | TA↑ |
|---|---|---|---|---|---|
| xyz+n | 68.83±0.95 | 0.7449±0.03 | 0.3749±0.02 | 2.6844±0.94 | 95.86±1.49 |
| xyz+n, aug | 71.06±0.65 | 0.7136±0.00 | 0.3820±0.01 | 1.1490±0.36 | 95.57±1.08 |
| xyz+rgb | **71.56**±0.41 | 0.7299±0.07 | 0.3666±0.04 | 1.3957±0.46 | 95.08±1.97 |
| xyz+rgb, aug | 69.35±0.95 | 1.0939±0.08 | 0.5678±0.05 | 1.7734±1.20 | 94.92±0.69 |
| xyz+rgb+n | 70.09±0.94 | **0.6192**±0.06 | **0.3078**±0.03 | 1.0330±0.47 | 94.31±1.99 |
| xyz+rgb+n, aug | 67.69±0.59 | 0.9274±0.06 | 0.4926±0.03 | **0.7215**±0.10 | **96.47**±1.94 |

| | Movable part | | | Motion type | | | Motion+Axis | | | Motion+Axis+Origin | | |
|---|---|---|---|---|---|---|---|---|---|---|---|---|
| Input | R | P | F1 | R | P | F1 | R | P | F1 | R | P | F1 |
| xyz+n | **18.82**±1.25 | 12.73±1.31 | 15.13±1.18 | **17.84**±1.32 | 6.00±0.74 | 8.95±0.92 | **17.46**±1.48 | 5.87±0.77 | 8.76±0.97 | **12.47**±0.82 | 4.19±0.44 | 6.24±0.54 |
| xyz+n, aug | 16.63±1.07 | **16.45**±1.88 | **16.50**±1.39 | 12.24±1.46 | **10.25**±2.11 | **11.11**±1.86 | 12.17±1.40 | **10.18**±2.05 | **11.04**±1.80 | 8.99±1.24 | **7.56**±1.71 | **8.18**±1.53 |
| xyz+rgb | 15.95±1.66 | 12.57±2.07 | 13.96±1.85 | 14.82±1.49 | 6.62±1.46 | 9.02±1.66 | 14.66±1.47 | 6.54±1.43 | 8.92±1.62 | 10.20±0.80 | 4.58±0.98 | 6.23±1.10 |
| xyz+rgb, aug | 15.27±1.18 | 15.80±1.86 | 15.47±1.40 | 11.34±0.23 | 10.06±0.66 | 10.63±0.29 | 11.19±0.33 | 9.91±0.57 | 10.48±0.21 | 5.97±0.20 | 5.33±0.54 | 5.61±0.37 |
| xyz+rgb+n | 15.50±0.53 | 12.37±0.36 | 13.76±0.43 | 14.06±0.73 | 7.53±0.36 | 9.79±0.38 | 13.98±0.66 | 7.49±0.34 | 9.74±0.33 | 10.28±0.53 | 5.50±0.18 | 7.15±0.19 |
| xyz+rgb+n, aug | 13.45±0.67 | 13.80±1.56 | 13.58±1.10 | 9.14±1.26 | 8.65±0.78 | 8.88±1.01 | 9.14±1.26 | 8.65±0.78 | 8.88±1.01 | 5.97±0.66 | 5.66±0.35 | 5.81±0.50 |

an NVIDIA TITAN X GPU. For our re-implementation, the MPN (stage 1) trains in about 3.5 hours for 100 epochs, and stage 2 and stage 3 training takes around 35 hours for 50 epochs.

We compare our re-implementation of Shape2Motion against the results reported in the original paper on the Shape2Motion dataset and show that our re-implementation gives comparable results (see Table 9.

**Ablation of inputs.** In addition to the position (xyz) point features, we experiment with using color (+rgb) and normal (+n) point features for the input point clouds. See Table 10 for a summary of comparisons between combinations of these different features for the input, as well as the use of data augmentation (aug).

**Detailed mobility estimation evaluation.** We compare mobility estimation performance of our Shape2Motion re-implementation and OPDPN using both ground truth object instance segmentations

Table 11: Mobility estimation on the val set comparing OPDPN [8] and Shape2Motion [16] on objects extracted using ground-truth (GT) and predicted segmentations from SSTNet [10]. We report both the mean and standard error across three random seeds.

| seg | method | IoU↑ | EPE↓ | MD↓ | OE↓ | TA↑ |
|---|---|---|---|---|---|---|
| GT | OPDPN | 54.80±2.67 | 0.96±0.25 | 0.48±0.12 | 3.99±3.02 | 92.59±7.41 |
| GT | S2M | **70.09**±0.94 | **0.62**±0.06 | **0.31**±0.03 | 1.03±0.47 | 94.31±1.99 |
| GT | S2M (aug) | 67.69±0.59 | 0.92±0.06 | 0.49±0.03 | **0.72**±0.10 | **96.47**±1.94 |
| SSTNet | S2M | **70.50**±1.62 | **0.59**±0.02 | **0.29**±0.01 | 1.41±0.80 | 94.85±1.80 |
| SSTNet | S2M (aug) | 69.22±0.52 | 1.04±0.23 | 0.52±0.12 | **0.57**±0.07 | 95.95±2.53 |

| | | Movable part | | | Motion type | | | Motion+Axis | | | Motion+Axis+Origin | | |
|---|---|---|---|---|---|---|---|---|---|---|---|---|---|
| seg | method | R | P | F1 | R | P | F1 | R | P | F1 | R | P | F1 |
| GT | OPDPN | 1.59±1.25 | 3.09±2.06 | 2.08±1.57 | 1.28±0.95 | 2.56±1.53 | 1.69±1.18 | 1.21±0.98 | 2.33±1.64 | 1.58±1.24 | 0.83±0.83 | 1.47±1.47 | 1.06±1.06 |
| GT | S2M | **15.50**±0.53 | 12.37±0.36 | **13.76**±0.43 | **14.06**±0.73 | 7.53±0.36 | **9.79**±0.38 | **13.98**±0.66 | 7.49±0.34 | **9.74**±0.33 | **10.28**±0.53 | 5.50±0.18 | **7.15**±0.19 |
| GT | S2M (aug) | 13.45±0.67 | **13.80**±1.56 | 13.58±1.10 | 9.14±1.26 | **8.65**±0.78 | 8.88±1.01 | 9.14±1.26 | **8.65**±0.78 | 8.88±1.01 | 5.97±0.66 | **5.66**±0.35 | 5.81±0.50 |
| SSTNet | S2M | **9.37**±0.72 | **17.02**±1.02 | **12.09**±0.85 | **8.69**±0.74 | **11.03**±0.14 | **9.68**±0.43 | **8.62**±0.73 | **10.94**±0.17 | **9.59**±0.41 | **6.05**±0.65 | **7.64**±0.18 | **6.72**±0.46 |
| SSTNet | S2M (aug) | 6.73±0.89 | 15.68±2.89 | 9.40±1.39 | 4.53±1.12 | 10.35±1.85 | 6.29±1.43 | 4.53±1.12 | 10.35±1.85 | 6.29±1.43 | 2.87±0.89 | 6.47±1.68 | 3.97±1.18 |

Table 12: Mobility estimation on the test set comparing OPDPN [8] and Shape2Motion [16] on objects extracted using ground-truth (GT) and predicted segmentations from SSTNet [10]. We report both the mean and standard error across three random seeds.

| seg | method | IoU↑ | EPE↓ | MD↓ | OE↓ | TA↑ |
|---|---|---|---|---|---|---|
| GT | OPDPN | 42.83±21.51 | **0.43**±0.22 | **0.22**±0.11 | 2.85±2.22 | 66.67±33.33 |
| GT | S2M | **75.15**±2.64 | 0.60±0.09 | 0.31±0.04 | **0.77**±0.04 | **99.12**±0.88 |
| GT | S2M (aug) | 72.96±2.38 | 0.50±0.10 | 0.27±0.05 | 1.58±0.90 | 96.55±1.75 |
| SSTNet | S2M | **75.08**±3.27 | 0.49±0.12 | **0.25**±0.06 | 0.96±0.04 | **100.00**±0.00 |
| SSTNet | S2M (aug) | 70.19±1.90 | 0.52±0.14 | 0.27±0.07 | **0.85**±0.08 | 98.33±1.67 |

| | | Movable part | | | Motion type | | | Motion+Axis | | | Motion+Axis+Origin | | |
|---|---|---|---|---|---|---|---|---|---|---|---|---|---|
| seg | method | R | P | F1 | R | P | F1 | R | P | F1 | R | P | F1 |
| GT | OPDPN | 1.52±0.98 | 3.14±1.79 | 2.04±1.26 | 1.52±0.98 | 3.14±1.79 | 2.04±1.26 | 1.42±0.88 | 2.95±1.64 | 1.90±1.14 | 0.91±0.63 | 1.85±1.15 | 1.22±0.82 |
| GT | S2M | **12.06**±0.10 | 10.13±0.30 | 11.00±0.15 | **11.15**±0.10 | 6.81±0.55 | **8.44**±0.43 | **11.15**±0.10 | 6.81±0.55 | **8.44**±0.43 | **8.41**±0.27 | 5.15±0.46 | **6.37**±0.39 |
| GT | S2M (aug) | **12.06**±2.84 | **12.56**±3.18 | **12.29**±3.00 | 8.81±2.53 | **8.06**±2.27 | 8.36±2.30 | 8.71±2.51 | **7.95**±2.20 | 8.25±2.25 | 6.28±1.67 | **5.68**±1.32 | 5.93±1.42 |
| SSTNet | S2M | 6.08±0.46 | 13.36±1.55 | 8.34±0.71 | **5.57**±0.51 | 9.78±2.44 | 6.97±0.91 | **5.57**±0.51 | 9.78±2.44 | 6.97±0.91 | **4.26**±0.18 | 7.46±1.75 | **5.33**±0.56 |
| SSTNet | S2M (aug) | **7.40**±0.97 | **20.21**±3.40 | **10.82**±1.53 | 5.37±1.24 | **15.57**±2.94 | **7.97**±1.75 | 5.37±1.24 | 15.57±2.94 | 7.97±1.75 | 3.55±0.54 | **10.34**±0.89 | 5.27±0.70 |

Table 13: Complete results including standard error for comparison of Shape2Motion [16] performance on open vs closed parts, evaluated on the val set with ground truth object instances.

| State | IoU↑ | EPE↓ | MD↓ | OE↓ | TA↑ |
|---|---|---|---|---|---|
| Closed | **67.76**±0.74 | 1.00±0.08 | 0.56±0.07 | 0.81±0.15 | 95.73±2.26 |
| Opened | 66.52±0.59 | **0.72**±0.00 | **0.36**±0.00 | **0.45**±0.09 | **100.00**±0.00 |

| | Movable part | | | Motion type | | | Motion+Axis | | | Motion+Axis+Origin | | |
|---|---|---|---|---|---|---|---|---|---|---|---|---|
| State | R | P | F1 | R | P | F1 | R | P | F1 | R | P | F1 |
| Closed | **14.16**±0.72 | **14.58**±1.34 | **14.34**±1.01 | **8.18**±1.54 | **7.77**±0.78 | **7.93**±1.17 | **8.18**±1.54 | **7.77**±0.78 | **7.93**±1.17 | **5.07**±0.92 | **4.82**±0.46 | **4.92**±0.70 |
| Opened | 12.51±1.19 | 9.57±1.77 | 10.77±1.59 | 6.26±1.19 | 3.74±0.56 | 4.67±0.78 | 6.26±1.19 | 3.74±0.56 | 4.67±0.78 | 4.75±1.08 | 2.83±0.56 | 3.54±0.74 |

and predicted segmentations from SSTNet. Tables 11 and 12 report the results on the val and test sets respectively. The val set table is repeated from the main paper, but here we provide standard error intervals on each mean reported for the metrics. We see that OPDPN is competitive in terms of performance as measured by the EPE, MD and TA metrics when ground truth object segmentations are used. In Table 13 we also analyze the performance difference between objects with open part state and objects with closed part state.

**Qualitative examples.** In Figure 17 we provide several examples comparing mobility estimation predictions using Shape2Motion (S2M) and OPDPN on inputs from the MultiScan val set. Simpler cases of objects with a single relatively large part (e.g., toilet lid in examples 1 and 2, and door in examples 3 and 4) are relatively easy. Overall, S2M provides more accurate motion axis and origin predictions. However, cases such as cabinet with drawers (example 5 and 6) and cabinetry with multiple doors opening in different directions (example 7 and 8) are quite challenging with missed part mobility predictions and high motion axis and origin errors.

|  | Input | GT | S2M | OPDPN |
|---|---|---|---|---|

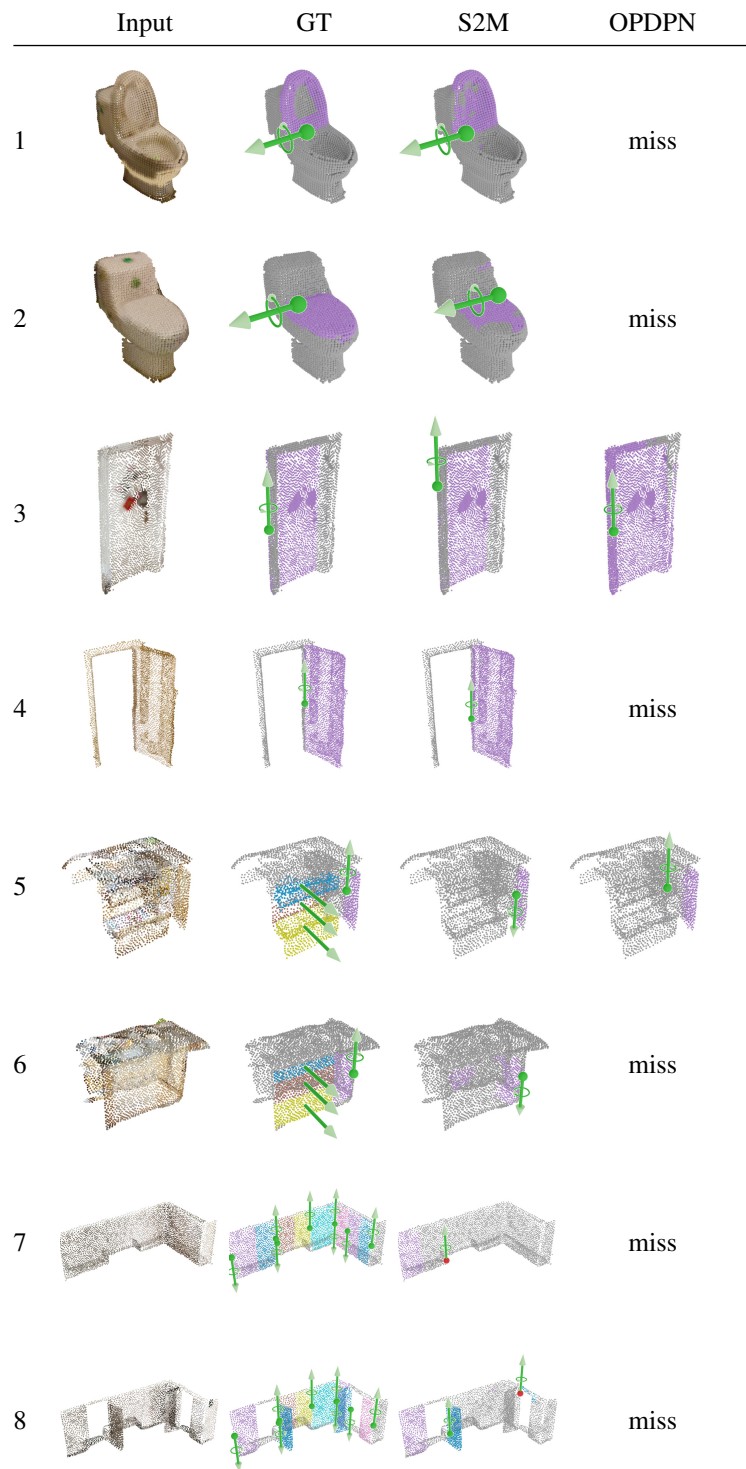

Figure 17: Examples of mobility estimation predictions on the val set using Shape2Motion (with aug) and OPDPN (no aug), both taking input point clouds with xyz+rgb+n features. For the ground truth (GT), the motion direction is shown with a green arrow and a circle around it indicating the rotation axis for rotations. For the predictions, predicted motion direction is shown in green if within $5°$ of GT, in orange if within $15°$ and in red if more than $15°$ off the GT direction. Shape2Motion (S2M) gives good predictions but sometimes the direction is 90 or 180 degrees off and the origin error is large (see examples 7, 8). OPDPN has low recall and often fails to predict any moving parts with IoU $\geq 0.5$.