# OpenReview forum: "MultiScan: Scalable RGBD scanning for 3D environments with articulated objects"
_NeurIPS.cc/2022/Conference — NeurIPS 2022 Accept_

### Official Review · Reviewer_bG7Q · 2022-07-09

**Rating:** 5
**Confidence:** 3
**Soundness:** 3 good
**Presentation:** 3 good
**Contribution:** 3 good

**Summary:**

This paper presents a new RGBD dataset called Multiscan focusing on articulated objects. When it is released, this dataset would provide raw RGBD data, camera poses, geometry reconstruction, and semantic annotation at the object level for each scan. The paper provides detailed explanation on data acquisition and annotation. It also provides benchmark on instance/part segmentation, and mobility prediction in the experiment section.

**Questions:**

- Scanning device and depth sensor consistency
The authors mentioned they developed both iOS and Android app for scanning, but in the paper, seems like only iOS devices are used. Is it correct? If both iOS and Android are used, presumably they use different depth sensors, then should make sure the same scene scanned by different devices should produce the consistent reconstruction.

**Strengths And Weaknesses:**

Strengths:
- The dataset especially with the articulate annotation is useful for the community.

- The code and data will be publicly available under permissive MIT and CC-BY-NC licenses. Since the capturing and annotation is quite efficient, other researchers could build upon the code to expand the dataset.

Weaknesses:
- Depth map resolution:
Compared to other RGBD datasets with a relatively high-resolution (e.g. at least 480*640) depth maps, the resolution in this dataset is much lower since it used mobile device for scanning. I suspect this could cause problems like losing details of object geometry when doing TSDF Fusion using low-res depth maps. Can the authors comment on this?
Another related comment: Although using mobile device for scanning can certainly help the scalability of the data acquisition, they often provides worse depth maps than dedicated depth sensors, such as Microsoft Azure. Did the authors compare the depth map quality between depth sensors when deciding the scanning device?

- Demo in supplementary material
It would be helpful if the author could provide an example of the scan and annotations.

- missing explanation on some details
At line 132, " We also filter out pixels where depth changes by more than 5cm between adjacent frames" Why is this useful and correct?
At line 135-136, how the voxel size and truncated distance are selected?

---

> ### Author Response · Authors · 2022-08-02
> **Response to R#bG7Q**
>
> **1. Depth map resolution: Compared to other RGBD datasets with a relatively high-resolution (e.g. at least 480*640) depth maps, the resolution in this dataset is much lower since it used mobile device for scanning. I suspect this could cause problems like losing details of object geometry when doing TSDF Fusion using low-res depth maps. Can the authors comment on this?**
>
> This is an excellent point. The hardware we used for our data collection (iPhone and iPad devices with built-in LiDAR sensor) provides 256x192 resolution depth which is lower than the nominal spatial resolution of sensors used by some prior work (Kinect v1 for NYUv2, Structure ST01 for ScanNet, and Kinect v2 or “Azure Kinect”).  These sensors can provide depth at 640x480 or higher nominal resolution.  However, the effective depth resolution is significantly lower and correlated with the resolution of the projected infrared (IR) fixed dot pattern for IR-based sensors (Kinect v1 and Structure ST01), or the use of pixel binning to improve range and range resolution for time-of-flight sensors (Kinect v2 / Azure Kinect).
>
> In early prototyping of our data collection workflow, we compared overall 3D reconstruction quality between the Kinect v1 / Structure sensors used for ScanNet and the built-in LiDAR on iOS devices. We found that the quality obtained with the latter, overall matched and sometimes even exceeded that obtained by depth sensors with higher nominal depth resolution.  We attribute this to improved depth accuracy and improved frame-to-frame tracking.  Moreover, the devices we used operate at a fairly high RGB resolution (1920x1440) and at high frequency (60Hz), enabling better tracking and high-resolution surface texture acquisition which is a significant reconstruction quality differentiator relative to prior work (see statistics comparing MultiScan with ARKitScenes in the common response to all reviewers).
>
> **2. Although using mobile device for scanning can certainly help the scalability of the data acquisition, they often provides worse depth maps than dedicated depth sensors, such as Microsoft Azure. Did the authors compare the depth map quality between depth sensors when deciding the scanning device?**
>
> We agree that using dedicated depth sensors can give higher quality depth maps that can improve reconstruction.  However, our focus is to develop a pipeline that can be used with commodity hardware available to many people.  We believe that research on real-world 3D scene representations is bottlenecked by the availability of 3D data and that by using sensors on mobile devices we can enable collection of more spaces.  Note that our pipeline for reconstruction and annotation can still be used by those with access to MS Azure and other dedicated depth sensors.
>
> **3. It would be helpful if the author could provide an example of the scan and annotations.**
>
> Please see the common response for a description of more examples of scan reconstruction quality available at this anonymous URL: [https://multiscan3d.github.io/](https://multiscan3d.github.io/) .
>
> **4. Missing explanation on some details**
>
> a. *At line 132, “We also filter out pixels where depth changes by more than 5cm between adjacent frames” Why is this useful and correct?* We found empirically that depth values that change rapidly between adjacent frames tend to be noisy.  When these values are filtered out, the resulting reconstruction is cleaner, with fewer floating artifacts (see supplement L49-54, and supplement Figure 2). We found that using depth confidence values alone was not sufficient for removing such floating artifacts.
>
> b. *At line 135-136, how the voxel size and truncated distance are selected?* We tried different voxel sizes and SDF truncation values on reconstructions of several sample scans.  We found that a truncation distance of 0.08m and voxel size of 9.77mm gave the best tradeoff between having sufficient resolution vs introducing noise.  More specifically, we started with the Open3D defaults and compared truncation values of 0.05m and 0.08m, and voxel sizes of 5.86mm, 9.77mm, 19.10mm and found that values of 0.08m and 9.77mm respectively worked best overall. Smaller voxel sizes for the TSDF volume provide higher resolution and can give more detailed surface reconstructions.  However, if the depth maps are not accurate, higher resolutions also introduce additional reconstruction noise.  A smaller truncation distance helps preserve more details in the scene, but is again vulnerable to depth noise.
>
> **5. Scanning device and depth sensor consistency. The authors mentioned they developed both iOS and Android app for scanning, but in the paper, seems like only iOS devices are used. Is it correct?**
>
> Yes, we only used iOS devices for data collection.  This was because we did not have access to Android devices with depth sensors.  We developed our pipeline to also support Android devices so that it is useful to a larger number of potential users.

---

> > ### Author Response · Authors · 2022-08-07
> > **Followup with R#bG7Q**
> >
> > Thank you R#bG7Q for your time and feedback. Please let us know if we can clarify any remaining questions or concerns.  In particular, we hope we have adequately addressed your questions about the scanning hardware and its impact on quality.  We are happy to provide additional information during the discussion period.

---

### Official Review · Reviewer_dEd3 · 2022-07-11

**Rating:** 4
**Confidence:** 3
**Soundness:** 3 good
**Presentation:** 3 good
**Contribution:** 2 fair

**Summary:**

The paper proposed a new dataset and annotation methodology for 3D scans with articulated objects. Unlike previous datasets such as ScanNet or ARKit scenes, this dataset includes part-level semantic segmentations part mobility parameters.  The authors create a specialized annotation tool for labels

**Questions:**

* Are the instance segmentations provided consistent across scans? What if there are multiple instances of the same object?
* Regarding the last point above, could the authors provide some clarification regarding the mobility prediction task?

**Limitations:**

Limitations are adequately discussed

**Strengths And Weaknesses:**

Strengths:
* This is the largest dynamic scans dataset to date. The authors propose an annotation pipeline which allows scans to be labeled and collected at scale.
* Scans are captured using common devices which reflect real use cases.
* I think this dataset could be useful to researchers working on dynamic reconstruction and part mobility prediction.

Weaknesses:
* It's difficult to assess the quality of the 3D scans in the paper and supplemental pdf.  From the visual examples provided in the paper, there appear to be of lower quality than other similar datasets such as ARKit scenes. Given that this is a dataset paper, some example meshes should have been included in the supplemental to better assess the quality of the reconstructions.
* The main body of paper should clearly state what annotations are provided with the dataset. All descriptions are of the mobility annotations are quite vague. The most informative descriptions I could are:
Ln 43: a dataset of densely annotated 3D interiors with object, part, and part mobility annotations
Ln 159: Object and part instances are correlated across scans of the same scene with consistent object and part IDs
Ln. 172: We define an articulated object to be an object consisting of rigid parts that are connected by joints

From these descriptions I am still not exactly what form of annotations are provided (i.e. object poses, axes of rotation, motion constraints?). Also what if the scene contains multiple instances of the same object (for example multiple instances of the same chair). In that case, it doesn't seem possible to match instances. I believe Rescan [9] solved this problem using a permutation matrix between object instances.

Basically, the Dataset (Sec 4) of the paper really needs to be expanded whereas some of the details in the acquisition and processing stages can be moved to the supplemental.

* Most of the related work section is focused on static datasets (e.g. ScanNet, ARKitScenes) whereas the overall goal of this dataset is more similar to Rescan [9]. I think more discussion of dynamic datasets is warranted.
* The motion parameter annotation methodology is not really a novel contribution as the authors note in the supplementary material that it is adopted from Xu et al.[47] with the addition of defined open/closed states.
* I wasn't able to completely understand the problem statement in 5.2 (Mobility prediction). From my understanding, the input is a point cloud Q at time t and the goal is to predict point cloud Q' at time t`. Is this problem even possible? If a door is opened, wouldn't it be equally probable that it (1) opens (2) remains half-open or (3) becomes fully open? I feel like I am missing something here if the authors could clarify that would be helpful.
* I am not really sure if there are any particular benefits of this dataset that could not be achieved by scaling up Rescan [9]. What sort of articulated objects captured by this approach would not be possible with the rescan methodology?

---

> ### Author Response · Authors · 2022-08-02
> **Response to R#dEd3**
>
> **1. Lack of examples of 3D scans to assess the quality of the reconstruction**
>
> Please see the common response for a description of more examples of scan reconstruction quality available at this anonymous URL: [https://multiscan3d.github.io/](https://multiscan3d.github.io/) .
>
> **2. Scans appear to be of lower quality than other similar datasets such as ARKit scenes**
>
> ARKitScenes [8] scans are PLY format mesh files with vertex colors only, unlike the textured mesh reconstructions we create in MultiScan. In addition, ARKitScenes does not provide semantic instance segmentations (only object bounding boxes). See the common response for some comparative statistics between ARKitScenes and MultiScan.
>
> **3. The main body of paper should clearly state what annotations are provided with the dataset**
>
> Thank you for the suggestion. We will move relevant information from the supplement (Section A.3, L95-97, L122-143) to the main paper.
>
> **4. Are the instance segmentations provided consistent across scans? What if there are multiple instances of the same object?**
>
> These are annotated as distinct object instances.  Note that the focus of our work is not to identify what objects in a scan are the “same object repeated”, but to connect objects in different temporal states.  Unlike Rescan [9] which has a proportionally large number of scenes with the same object repeated multiple types (same style of chair), our scenes are mostly taken from home environments where this occurs less frequently.  Our focus is also more on articulated objects such as kitchen cabinets and their changing states. While instances of such objects can be similar to one another, they are typically fixed in place and cannot be moved from one position to another.
>
> **5. Most of the related work section is focused on static datasets**
>
> Please see L64-77 for a discussion of “Interactive environments and objects” which includes datasets of articulated objects and CAD scenes, and L78-93 for a discussion of “Reconstruction of articulated objects”.  If there are any additional dynamic datasets that should be discussed, please let us know.
>
> **6. “I wasn't able to completely understand the problem statement in 5.2 (Mobility prediction).  From my understanding, the input is a point cloud Q at time t and the goal is to predict point cloud Q' at time t`. Is this problem even possible?”**
>
> This is a misunderstanding.  That is not our mobility prediction problem statement.  We set up mobility prediction similarly to prior work, where the input is a point cloud Q and the goal is to predict what parts can move and their motion parameters in 3D.  Specifically, we predict a set of moving parts with their motion types, motion axis direction, and origin for rotational joints (see L257-260).  With this predicted information, it is possible to convert the point cloud Q into a dynamic point cloud (e.g., the cabinet door can be opened/closed by taking the points associated with the cabinet door and rotating them about the motion axis).  However, we do not attempt to predict a different point cloud Q’ at time t’.
>
> **7. What sort of articulated objects captured by this approach would not be possible with the Rescan methodology?**
>
> Our annotation pipeline allows for the annotation of parts and part mobility information, which was not possible with the Rescan [9] annotation pipeline.  Also note that similarly to ARKitScenes, Rescan does not provide textured mesh reconstructions that can capture finer details which are useful in annotating object parts.

---

> > ### Author Response · Authors · 2022-08-07
> > **Followup with R#dEd3**
> >
> > Thank you R#dEd3 for your time and feedback. Please let us know if we can clarify any remaining questions or concerns.  In particular, we hope we have adequately addressed your questions about dataset and annotation quality, the mobility prediction task definition, and differences compared to Rescan.  We are happy to provide additional information during the discussion period.

---

### Official Review · Reviewer_gq65 · 2022-07-12

**Rating:** 4
**Confidence:** 4
**Soundness:** 3 good
**Presentation:** 3 good
**Contribution:** 2 fair

**Summary:**

This paper presents MultiScan, an RGBD dataset for indoor scenes with semantically annotated 3D objects. The acquisition approach is similar to ScanNet [7] which used many users with commodity iOS and Android devices with active LiDAR sensors. Compared to existing 3D datasets, this paper achieved dense textured meshes, multi-level hierarchical annotations (needs elaboration) and object parts correspondences with respect to time and motion changes.

While the paper presented an interesting extension to existing large scale datasets, I think this paper is more suited for the NeurIPS Datasets and Benchmark track.

**Questions:**

None.

**Limitations:**

I think the limitations have been addressed.

**Strengths And Weaknesses:**

The strength of this paper is that it presented a large scale dataset with a comprehensive pipeline for annotations. It also has advantages for recording scenes at multiple timestamps with object motions such as opening a drawer/window.

For the weaknesses, the paper does not present the details for the hierarchical part labels and how it can be evaluated/benchmarked. It is also not clear how the hierarchical annotation is different from ScanNet [7].

---

> ### Author Response · Authors · 2022-08-02
> **Response to R#gq65**
>
> **1. Details of the hierarchical part labels**
>
> The hierarchical part labels are explained in the supplement L122-124  (*“Annotators provide a label of the form `object_id:part_id = object_category.object_index:part_category.part_index` that is used to identify the object and part category and instance.”*).  For instance, we annotate a cabinet with two openable doors as having: one static part `cabinet.1:cabinet.1` and two doors `cabinet.1:door.1` and `cabinet.1:door.2`.  Each of the two doors will also have annotated motion parameters (see common response).
>
> **2. How can the hierarchical part labels be used for benchmarks?**
>
> In our submission, we take the hierarchical part labels and construct three benchmark experiments on different levels of the hierarchy (see Section 5.1).
> 1. *Object instance segmentation given the entire scene.*  For this task, all part labels belonging to the same object are aggregated into a single object instance (e.g., the three cabinet part labels are combined into one object label `cabinet.1`).  See Figure 1 third column and Table 2.
> 2. *Part instance segmentation given ground truth object segmentation.*  Having the hierarchical annotation allows us to extract individual objects from the scene and construct a benchmark that focuses on the part instance segmentation given the object segmentation. See Figure 1 fourth column and Table 3 left.
> 3. *Part instance segmentation given predicted object segmentations.*  As we have both the object and part level annotations we then create a combined benchmark where we investigate the performance of part-level instance segmentation at the scene level.  We take a two-stage approach where we first predict the objects and then for each predicted object, we perform part-level segmentation.  See Table 3 right.  As expected, this approach results in lower performance than when ground-truth object segmentation is provided.
> These three experiments provide initial benchmarks with the MultiScan object-part hierarchy.  The MultiScan data can allow for the development of additional tasks (each with their appropriate evaluation metrics).  For instance, we also benchmark part mobility prediction for each object (given ground-truth object segmentations) on our dataset.
>
> **3. How is the hierarchical annotation different from ScanNet?**
>
> ScanNet [7] does not provide hierarchical annotation of objects and their parts.  Please see common response for a detailed list of the annotations that MultiScan provides that are not in ScanNet.

---

> > ### Author Response · Authors · 2022-08-07
> > **Followup with R#gq65**
> >
> > Thank you R#gq65 for your time and feedback. Please let us know if we can clarify any remaining questions or concerns.  In particular, we hope we have adequately addressed your questions about the hierarchical part labeling and differences compared to ScanNet.  We are happy to provide additional information during the discussion period.

---

> > ### Comment · Reviewer_gq65 · 2022-08-09
> > **My rating remains the same.**
> >
> > Thanks to the authors to spend time addressing my concerns. I think this paper should be submitted to the dataset track, not the main program. My rating remains as reject.

---

> > > ### Author Response · Authors · 2022-08-09
> > > **Response to R#gq65**
> > >
> > > We thank the reviewer for their effort in reviewing our paper and recognize the reviewer's opinion.  However, we would like to point out that the NeurIPS call for paper explicitly lists "Infrastructure (e.g., datasets, competitions, implementations, libraries)" as one of the paper topics  sought by the main track of NeurIPS (see https://neurips.cc/Conferences/2022/CallForPapers).  In our opinion, the existence of a parallel datasets and benchmarks track should not be the sole grounds for rejection of otherwise legitimate work.  We trust that the reviewers and area chair will interpret the NeurIPS call for papers and associated policies without taking this restrictive view which can in our opinion be quite harmful for the NeurIPS community as a whole and disincetivize dataset and benchmark contributions at the main track of NeurIPS.

---

### Author Response · Authors · 2022-08-02
**Common response to all reviewers**

We thank all reviewers for their time and thoughtful feedback.  The reviewers noted that we contribute a “large scale dataset with a comprehensive pipeline for annotations” (R#gq65), that MultiScan is the “largest dynamic scans dataset to date” (R#dEd3) and is “useful to researchers working on dynamic reconstruction and part mobility prediction” (R#dEd3). Furthermore, R#bG7Q states that “articulation annotation is useful for the community” and that the “efficient capturing and annotation pipeline” enables “other researchers could build upon the code to expand the dataset”.  In addition to the scalable acquisition and annotation pipeline and large-scale articulated scan dataset the reviewers have noted, we also carried out a systematic benchmark of methods for part instance segmentation and part mobility parameter estimation, laying foundations for future work on these challenging tasks.

Here, we address questions that are common across reviewers.  We also provide responses to specific reviewer questions directly below each review.

**Novelty of MultiScan relative to prior datasets**

MultiScan is the first dataset of articulated, interactive scans of indoor scenes.  While there have been prior efforts on articulated single object datasets (both synthetic CAD models and scanned real objects), we provide the first dataset of indoor scenes annotated with movable object parts and their motion parameters.  Several annotations are unique to MultiScan:
1. Semantic instance segmentations for both objects and their parts
2. Semantically meaningful oriented bounding boxes (OBBs) with consistently defined front and up orientations for every object
3. Annotation of object parts that can move and how they move, with motion parameters including motion type (revolute vs prismatic), motion axis and origin, and a semantically meaningful motion range (fully closed to fully open)

We provide the details of the annotation interface and process in the supplement (Section A.3).  Also see Table 1 and Section 2 in the main paper for more details on how Multiscan differs from related efforts.

**Scan and annotation quality is hard to judge**

R#dEd3 mentioned it is “difficult to assess the quality of the 3D scans” and R#bG7Q that it would be “helpful if the author could provide an example of the scan and annotations.”  We provide higher-resolution animations and interactive 3D mesh visualizations of example scans and annotations from MultiScan at this anonymized URL: [https://multiscan3d.github.io/](https://multiscan3d.github.io/) The page includes all examples from Fig 11 of the supplement as well as additional scans.

Note that our annotation pipeline can also be applied to earlier datasets (ScanNet [7], ARKitScenes [8], Rescan [9], RIO [41]) for part and part mobility annotation.  However, these datasets lack textured mesh reconstructions and fine-grained (i.e. at the level of mesh triangles) semantic instance annotations.  In MultiScan, we use textured mesh reconstructions and triangle-level semantic annotation to capture finer details that are important for part segmentation.  R#dEd3 specifically inquired about MultiScan reconstruction quality relative to ARKitScenes.  Unlike MultiScan, ARKitScenes reconstructions are vertex-colored PLY format meshes that do not use texture maps for fine-grained surface detail.  Thus, ARKitScenes scans are limited by the geometric resolution of the mesh and often miss or “blur out” details such as handles, knobs and cabinetry edges etc. which are particularly important for moving part annotation.  To give a general sense of the surface detail captured in our reconstructions we compare the vertex-based color resolution of ARKitScenes against the texture-based color resolution of MultiScan, in terms of mean number of color values per mesh surface area unit. ARKitScenes has $0.658\frac{\text{vertices}}{\text{cm}^2}$, whereas MultiScan has $79.5 \frac{\text{texels}}{\text{cm}^2}$ (a more than 100x difference in reconstructed surface color value resolution).

We are happy to provide further information to clarify any additional questions by the reviewers.

---

> ### Author Response · Authors · 2022-08-07
> **Followup with all reviewers**
>
> We thank all the reviewers once again for their time and feedback.  Please don’t hesitate to follow up if you have any remaining questions or concerns that we can clarify during the discussion period.

---

### Meta-Review · Area_Chair_TQ1T · 2022-08-27

**Recommendation:** Accept
**Confidence:** Certain

**Metareview:**

The reviewers tend to agree on the value of this 3D dataset, but point to some questions about labelling and accuracy.  The rebuttal very convincingly addresses these points, clarifying the novelty and value of this new dataset.

I agree with the authors that datasets are clearly in scope for the main NeurIPS program and that the datasets track explicitly includes as a FAQ:   "My work is in scope for this track but possibly also for the main conference. Where should I submit it?" with the answer "This is ultimately your choice".


**Award:**

No

---

### Decision · Program_Chairs · 2022-09-14

Accept